# Cosmopolitan no more: Phylogenetics and reproductive mode reveal a global species complex in a marine mollusk (Teredinidae)

Nancy C. Treneman[1]*, Kelli L. DeLeon[2], J. Reuben Shipway[3,4], Luísa M. S. Borges[4,5], Kenneth A. Hayes[6]

1 Oregon Institute of Marine Biology, Charleston, Oregon, United States of America, 2 Florida Museum of Natural History, Gainesville, Florida, United States of America, 3 Naked Clam Limited, Cambridge, United Kingdom, 4 International Marine Wood-Borer Network (IMWBN), Global Research Network, Birmingham, United Kingdom, 5 L³ Scientific Solutions, Geesthacht, Germany, 6 Hawai'i Biological Survey, Bernice P. Bishop Museum, Honolulu, Hawaii, United States of America

* ntreneman@gmail.com

## Abstract

Quantifying biodiversity is challenging when morphology is conserved in taxa with extensive geographic distributions generated in part by human activities. Shipworms, xylophagous wood-boring clams, have been dispersed throughout the world's oceans by wooden vessels, aquaculture equipment, and in ballast water. Consequently, many species are considered cosmopolitan, with their geographic origin obscured by their extensive distribution. Several cryptic species pairs possessing different reproductive modes are known in the Teredinidae. However, the genetic, ecological, and geographic relationships within these pairs remain unexplored. Members of the *Lyrodus pedicellatus* complex, both long- and short-term brooders, are found on coastlines of five continents. Phylogenetic, anatomical, ecological, and geographic data were collected on shipworms extracted from test panels, fixed submerged natural wood and driftwood, from multiple locations, primarily in the Hawaiian Archipelago, a center of wooden vessel traffic since the 1400s. Phylogenetic analysis, using multiple loci of *L. pedicellatus* from Hawai'i, Europe, the Mediterranean, Japan, Florida (USA), and California (USA), revealed seven genetically distinct cryptic species comprised of short- and long-term brooders. Reproductive mode was determined to be an inherited trait within the species in this study. Herein we discuss these findings and describe a new member of this complex, *Lyrodus reginae* sp. nov., from Hawai'i. Historically, *L. pedicellatus* was considered a cosmopolitan species. Our integrative approach reveals a more complicated story, indicating the evolution of multiple cryptic species with different reproductive strategies.

**Data availability statement:** All relevant data are within the manuscript.

**Funding:** Partial Funding for this research was provided by the Charles H. and Margaret B. Edmondson Research Grant in Aide, University of Hawaii at Manoa, awarded through the Bernice P. Bishop Museum, received by NC Treneman, yearly from 2017 to 2022. https://manoa.hawaii.edu/lifesciences/graduate/zoology-graduate-program/zoology-graduate-student-research-awards/ The funders had no role in study design, data collection and analysis, decision to publish, or preparation of the manuscript.

**Competing interests:** The authors have declared that no competing interests exist.

## Introduction

The wood-boring bivalves of the family Teredinidae, commonly known as shipworms, are found throughout the world's oceans [1]. The majority of teredinids bore into and feed on wood, except for a few species inhabiting seagrass rhizomes, mud, and limestone [2–4]. Although vilified as pests for their destruction of wooden ships and maritime structures [5,6], shipworms are keystone ecosystem engineers and crucial contributors to the ocean energy budget via the conversion of wood to marine biomass [7,8]. Research on the diversity and biology of shipworms, essential to an understanding of marine ecosystems, is hindered by their challenging taxonomy and residence in wood, making them difficult to identify and study *in situ*. Field survey methods, specific for woodborers, and the use of molecular and new imaging techniques have facilitated the recent discovery of new species and cryptic species complexes within the family [9–12].

Shell morphology, the primary taxonomic diagnostic character in most bivalves, is uninformative in shipworms as no consistent differences have been identified between the shells of species and even genera, with a few notable exceptions [2]. Gross soft tissue anatomy presents the same problem, except in a few instances, as in *Neoteredo reynei* (Bartsch, 1922), where the distal mantle has two lateral wing-like lappets [1,13]. The most useful diagnostic character in shipworms is pallet morphology [1,14]. These structures are positioned lateral to the siphons and seal the tunnel entrance, protecting the animal from predation and environmental stressors. Pallets have a calcium carbonate stalk with distal blades and exhibit a stunning diversity of shapes and sizes, from the paddle-shaped pallets of *Teredora* to the feather-like pallets in *Bankia* [1]. The pallets of the genus *Lyrodus* have a blade topped by a periostracal cap at the distal end [10]. The structure of the periostracal cap and shape of the blade are key elements for species identification within this genus.

Shipworm taxonomy is complicated by cryptic diversity, where the pallets of genetically isolated species are indistinguishable. Several of these species complexes are comprised of cryptic species with different reproductive modes [15]. Four forms of reproduction are found in the Teredinidae: oviparous, short-term brooding (STB), synchronous long-term brooding (SyLTB), and sequential long-term brooding (SqLTB) [15–17]. In STBs, larvae of the same age class are brooded in the gills and move freely within the gill lamellae. STBs spawn their larvae simultaneously at the straight-hinge veliger stage (D-stage). Long-term brooders (LTB) spawn larvae at the pediveliger stage. There are two forms of LTB in the Teredinidae: synchronous (SyLTB) and sequential (SqLTB) [15,17]. Brooded larvae in SyLTBs belong to a single age class, move freely within the gill lamellae, and are released en masse during discrete spawning events. The majority of LTB shipworms are SyLTBs [15,17]. In contrast, the gills of SqLTBs contain a continuum of age classes. Larvae in SqLTBs reside in 'closed pouches' within the gill lamellae, and are released progressively as they mature to the pediveliger stage [15,17].

The first cryptic species discovered in the Teredinidae was a STB, *Lyrodus floridanus* (Bartsch, 1922) (type locality: Tampa, Florida, USA) [16]. Calloway and Turner [16] collected STB shipworms with pallet morphology identical to that of

*L. pedicellatus*, a SqLTB, and resurrected the name *L. floridanus* based on collection locality. Recently, Borges and Merckelbach [10] used molecular diagnostic characters derived from DNA sequences to describe another cryptic species in the *L. pedicellatus* complex, *L. mersinensis* Borges & Merckelbach, 2018.

Turner [1] synonymized several hundred shipworm species, reducing the valid taxa to 68, consequently making several species appear to be cosmopolitan in distribution. The subsequent discovery of cryptic taxa in the Teredinidae [10,11,16] highlights the need for a reassessment of shipworm diversity. This is especially true in locations, such as Hawai'i, with a long history of maritime traffic, due to the human-mediated dispersal of shipworms around the world in the hulls of wooden vessels, ballast water, and aquacultural materials [18,19]. The unique location of the Hawaiian Archipelago, isolated by >3,200 km of ocean from the nearest continent, makes it a useful port of call, and vessels from all over the world came, and still come, to Hawai'i [20]. In addition, shipworms are carried in driftwood to Hawaiian shores, arriving from both the east and west sides of the Pacific [14,21]. These anthropogenic and natural dispersal mechanisms make it possible to find introduced, and possibly endemic, shipworm species in Hawai'i.

The first surveys of the Teredinidae in Hawai'i by P. Bartsch in 1907 and 1920 [*fide* 22, *fide* 23,24] occurred long after the arrival of Polynesian canoes (1000–1100 A.D.) and European wooden vessels (1778 A.D.) [20,25,26]. The most comprehensive surveys of the Hawaiian Archipelago were carried out by Edmondson [27–29]. During 1939–1940, Edmondson deployed wooden test panels and collected driftwood throughout the archipelago, including Midway Atoll. From this survey and others in subsequent years, Edmondson determined there were 12 species of shipworm resident in Hawai'i, including *L. pedicellatus*. The Clapp Laboratories, in their wide-ranging effort to determine shipworm species distributions, set out test panels at Midway and Pearl Harbor, O'ahu from 1944–1959. Their efforts discovered nine species, representing a subset of Edmondson's species [30].

Herein we report our findings on STB and SqLTB members of the *Lyrodus pedicellatus* species complex collected during surveys in Hawai'i, California, and Japan. With a combination of morphology, reproductive biology, DNA sequencing, and examination of archival material, we discovered seven cryptic species within the *L. pedicellatus* complex, including *Lyrodus reginae* sp. nov. described herein. Larval brooding strategies, morphology, and ecology of these species are discussed.

## Materials and methods

Test panels (n = 138 of Douglas fir and oak, 25 x 18.5 x 2 cm) were deployed on O'ahu, Hawai'i Island, and Midway Atoll at a total of 28 sites from 2015 to 2025. During the same period, driftwood (dry and submerged) and natural fixed submerged wood (e.g., submerged and attached tree branches and roots, Fig 1) were collected on O'ahu, Hawai'i Is., Kaua'i, Kure Atoll, and Midway Atoll across 70 locations. Location, habitat, depth, water temperature and salinity were recorded. Submerged driftwood was collected at two sites in 2019 in San Francisco Bay, California (37.572263, −122.27917 and 37.83227, −122.474591). Test panels were deployed at Amakusa, Japan (n = 6; 2 pine, 2 Douglas fir, 2 Oak) (32.527459, 130.034636) from 2017 to 2018.

Wood was generally processed within two days of collection. When processing was delayed, the wood was frozen or preserved in 95% ethanol for later processing. Woodworking tools were used to extract shipworms. Body length, anatomy, reproductive status, and behavior such as spawning, siphon and pallet activity were assessed before fixation in 95% ethanol. Brooding mode was determined by observing larval stages in the gill cavity in hand and with a stereoscope, live spawning by specimens *in situ* (undisturbed within their tunnels) and in living, undamaged individuals removed from their tunnels. Gill lamellae sections were examined with a compound scope (Leitz Labourlux D) for larval stages. Siphons and/or mantle tissue were taken from representative specimens and preserved in 95% ethanol for DNA extraction. The remaining bodies (vouchers), in 95% ethanol, are retained at the Pacific Center for Molecular Biodiversity (PCMB) at the Bernice P. Bishop Museum (BPBM), Honolulu, Hawai'i. After fixation specimens deposited in museum collections were transferred to 80% ethanol. The holotype and paratype of *L. reginae* sp. nov. are deposited at the BPBM. The remaining specimens are in collections at the BPBM, Museum of Comparative Zoology (MCZ), and NCT's personal collection.

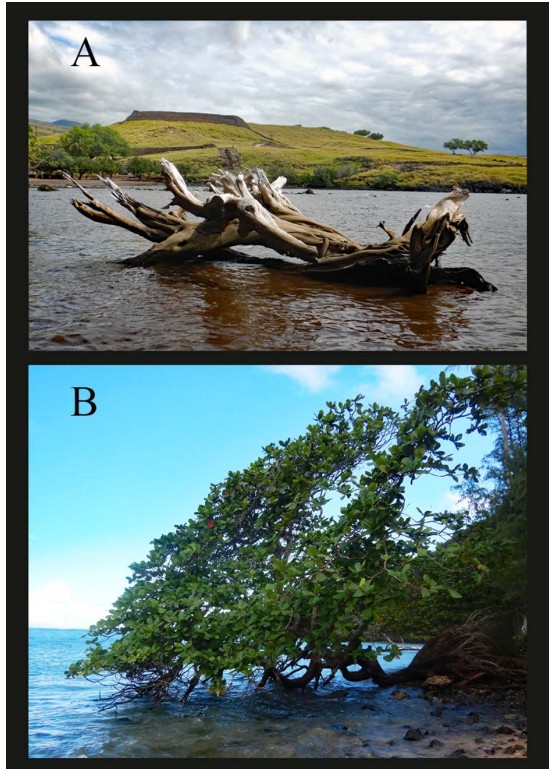

**Fig 1. Natural fixed submerged wood examples.** Definition: Wood items attached to the shore and underwater, such as tree branches attached to the tree and dipping into the water, or partially submerged logs in place for a number of years. **(A)** Mesquite log, at low tide, in place at least 2.5 years, Pelekane Estuary, Hawaiʻi Island. **(B)** Indian Almond tree, Kahana Bay, Oʻahu.

All necessary permits were obtained for the described study, which complied with all relevant regulations. Permits were not required for the majority of locations, as research was conducted and coordinated with state and federal agencies and staff of the U.S. Navy, U.S. Fish and Wildlife, State of Hawaiʻi, the National Oceanic and Atmospheric Administration (NOAA), and the Amakusa Marine Station (Japan). Shipworms are not a protected taxa and are considered invasive in Hawaiʻi. Permits were required for two locations and obtained from the State of California Fish and Wildlife and the Oʻahu National Wildlife Refuge Complex.

### Electron microscopy

Larvae were dissected from the gills of shipworms, dehydrated in absolute ethanol, and dried using hexamethyldisilizane. Specimens were mounted on aluminum stubs and coated in gold to a thickness of 20 nm. Larvae were then visualized on the JEOL 6610 LV SEM at the Plymouth Electron Microscopy Centre at a working distance of 10 mm and imaged at 15 kV.

### Comparative material

Specimens of the *L. pedicellatus* clade, including the type material, were examined at the following museums: Bernice P. Bishop Museum (BPBM, Honolulu, Hawaiʻi, USA), California Academy of Sciences (CAS, San Francisco, California, USA), Museum für Naturkunde (ZMB, Berlin, Germany), Museum of Comparative Zoology (MCZ, Cambridge, Massachusetts, USA), Muséum National d'Histoire Naturelle (MNHN, Paris, France), the National Museum of Nature and Science,

Tokyo (NMNS, Tsukuba, Japan; NMNS PMR, Department of Geology and Paleontology; NSMT-Mo, Department of Zoology), and the Smithsonian National Museum of Natural History (USNM, Washington, D.C., USA).

## DNA extraction, sequencing, and phylogenetic analysis

Total genomic DNA (gDNA) was extracted from an approximately 1 mm³ piece of siphon tissue using the Macherey-Nagel NucleoSpin® Tissue Kit following the manufacturer's instructions, with the exception that the final elution was with 60 µl of elution buffer supplied with the kit, and gDNA stored at −20 °C prior to amplification via polymerase chain reaction (PCR). Portions of two mitochondrial genes, *16S rRNA* and cytochrome *c* oxidase subunit I (*COI*), and fragments of two nuclear-encoded ribosomal genes, *18S rRNA* and *28S rRNA,* were amplified using primers listed in Table 1. Reactions were carried out in 25 µl volumes containing 1–2 µl template DNA and a final concentration of 1 U of MangoTaq™ DNA polymerase (Bioline), 1× reaction buffer, 0.2 mM each dNTP, 2.5 mM MgCl$_2$ and 0.75 µM of each primer, 10 µg BSA, and 0.5% DMSO. Cycling parameters were one cycle of 5 min at 95 °C, 1 min at 45–50 °C, 2 min at 72 °C, followed by 35–40 cycles of 95 °C, 48–52 °C, and 72 °C for 30 s each, and a final extension of 5 min at 72 °C. A final 4 °C incubation of 30 min terminated each reaction. The amount and specificity of amplifications were verified via agarose electrophoresis and single product amplicons were cycle sequenced using the ABI BigDye terminator kits (Perkin-Elmer Applied Biosystems, Inc.). Sequence electrophoreses and analyses were carried out on an ABI 3730XL (Perkin-Elmer Applied Biosystems, Inc.) at Eurofins Genomics, LLC. All loci were initially sequenced in one direction, and all unique haplotypes sequenced in both directions. The *COI* fragment was sequenced for all individuals, and subsets of these were selected based on unique *COI* haplotypes and sequenced for *16S, 18S,* and *28S*. Due to lower variability in the other three loci, not all individuals with a unique *COI* haplotype were sequenced for all other loci. All sequences have been uploaded to the Barcode of Life Data Systems (BOLD).

Electropherograms were checked for errors, edited, and assembled using Geneious Prime 2023 (http://www.geneious.com/). Sequences of *COI* were unambiguously aligned using MAFFT ver. 7.388 with the iterative refinement method E-INS-I [39] implemented in Geneious Prime. Alignments were checked against amino acid sequences as references. Ribosomal genes were aligned using MAFFT and refined using Gblocks ver. 0.91b [40] to remove regions of ambiguous homology created by the addition of gaps during initial alignment and the hypervariable nature of some ribosomal regions. Phylogenetic analyses were done with and without these regions to evaluate their impact. Sequence alignments were concatenated in Geneious Prime and exported as fasta files for phylogenetic analysis.

Phylogenetic reconstruction was conducted using maximum likelihood (ML) in IQ-TREE ver. 1.6.12 [41] and Bayesian Inference (BI) via MrBayes, [42] both implemented in PhyloSuite ver. 1.2.3 [43,44]. The best-fit partitioning scheme and the most appropriate substitution model for each partition were estimated using the integrated PartitionFinder2 ver. 2.1.1 [45]. Maximum likelihood nodal support values were estimated with 20,000 ultrafast bootstrap replicates [46]. MrBayes

**Table 1. Primers used for amplification and sequencing for this study.**

| Locus | T$_A$ °C | Primers F/R (Reference) |
|---|---|---|
| *16S* | 48-50 | 16Sar/16Sbr [31] |
| *16S* | 48-50 | 16Sar/16S2 [31,32] |
| *COI* | 45-48 | Mat-fw/Mat-rev [33] |
| *COI* | Seq only | Ter-fw II [34] |
| *COI* | 45-48 | Ter-fw III/Ter-rev I [34] |
| *28S* | 48-50 | NLF 184–21/28S-1600R [35,36] |
| *18S* | 50-52 | 18Sfw/18Srev [37] |
| *18S* | 50-52 | 18F509/18Srev [37,38] |

T$_A$ = annealing temperature for PCR; Seq only = primer was only used for sequencing.

analyses consisted of four parallel runs with four chains of eight million generations each. Trees were sampled every 1,000 generations with a burn-in of 25%. Nodal support was assessed with posterior probabilities and effective sample size and convergence diagnostics reviewed after each run. Genetic distances were calculated using a K2P model implemented in MEGA 11 [47] and DeSigNate [48] was used to detect and identify signature molecular characters for all markers and the sequence positions diagnostic among species in the *L. pedicellatus* complex included in this study. The nuclear ribosomal loci, *18S* and *28S*, were used to help resolve relationships in the phylogenetic analyses, but the unique character states for these loci should be interpreted cautiously as diagnostic characters within the *L. pedicellatus* complex given the limited depth of taxon sampling. PCMB catalog numbers for specimens used in genetic analysis along with sequences obtained from GenBank are provided in Table 2.

### Species delimitation

Species delimitation was done using phylogenetic criteria under the Unified Species Concept [51,52]. Two methods were used to delineate species based on mitochondrial *COI* sequences without *a priori* group designations: Assemble Species by Automatic Partitioning (ASAP: [53]) using the Jukes-Cantor (JC69) substitution model and Bayesian implementation of the Poisson tree processes model (bPTP: [54]). The input for the bPTP analysis was the maximum likelihood tree estimated based on *COI* sequences in IQ-Tree. Candidate species partitions were ranked by ASAP score, with lower scores indicating better support. The lowest-ranked partition was identified and evaluated as our primary species hypothesis in the context of phylogenetic structure, molecular diagnostic characters, and reproductive mode; alternative high-ranking partitions were examined to assess partition stability. Similarly, bPTP delimited entities receiving relatively high posterior probability support were treated as candidate species hypotheses and evaluated for congruence with phylogenetic structure, molecular diagnostic characters, and reproductive mode. Together, outputs from ASAP and bPTP were treated as exploratory species hypotheses rather than as determinative criteria. Putative species partitions were considered supported only when they were congruent with phylogenetic structure inferred from multilocus analyses, the presence of unique combinations of molecular diagnostic characters, and consistent reproductive mode within clades. No fixed genetic distance thresholds or posterior probability cutoffs were applied; instead, delimitation decisions followed a congruence-based framework under the Unified Species Concept. Genetic divergence values were evaluated solely as comparative metrics in conjunction with phylogenetic structure, reproductive biology, and geographic evidence, consistent with modern integrative taxonomic frameworks.

### Nomenclatural acts

The electronic edition of this article conforms to the requirements of the amended International Code of Zoological Nomenclature, and hence the new names contained herein are available under that Code from the electronic edition of this article. This published work and the nomenclatural acts it contains have been registered in ZooBank, the online registration system for the ICZN. The ZooBank LSIDs (Life Science Identifiers) can be resolved and the associated information viewed through any standard web browser by appending the LSID to the prefix "http://zoobank.org/". The LSID for this publication is: urn:lsid:zoobank.org:pub:9B03F99E-7275-4A40-B299-200A4C0BABD1.The electronic edition of this work was published in a journal with an ISSN, and has been archived and is available from the following digital repositories: PubMed Central and LOCKSS.

### Results

From 2016 to 2025, a total of 1,677 living specimens were collected in Hawaiʻi, 295 with the same pallet morphology as *Lyrodus pedicellatus,* 132 of these brooding larvae. Eighteen of the latter were SqLTBs, verified by the presence of pediveliger larvae and successively smaller larvae in sequence within the gill lamellae. This SqLTB was collected at five sites: (Fig 2) in natural fixed wood at two sites on the west side of Hawaiʻi Island, and in test panels on Oʻahu: Pearl Harbor (2 sites), and Waiʻanae Harbor – the only site this species was consistently present (4/10 test panel deployments).

**Table 2. Specimen details for samples included in genetic analysis.**

| ID | Specimen | Locality | Reference | BOLD or GenBank Numbers | COI | 16S | 18S | 28S |
|---|---|---|---|---|---|---|---|---|
| *Bankia bipalmulata* | PCMB50792, MAL032267 | Keʻehi H, Oʻahu, HI, USA; TP, 1.3 m; 21.317445°, −157.892051° | this study | LREG035−23 | X | X | X | X |
| *Bankia bipalmulata* | PCMB51880, MAL032294 | Honolulu H, Oʻahu, HI, USA; TP, 2 m; 21.303865°, −157.870435° | this study | LREG036−23 | X | X | | |
| *Bankia bipalmulata* | PCMB51890, MAL032304 | Waiʻanae H, Oʻahu, HI, USA; TP, 1 m; 21.449683°, −158.197700° | this study | LREG038−25 | X | X | | |
| *Lyrodus affinis* | PCMB50802, MAL032277 | Keālia Bh, Kauaʻi, HI, USA; Df, 1m; 22.090319°, −159.304898° | this study | LREG004−23 | X | X | | |
| *Lyrodus affinis* | PCMB50807, MAL032282 | Nāwiliwili H, Kauaʻi, HI, USA; Df, 1 m; 21.949860°, −159.357818° | this study | LREG005−23 | X | X | X | |
| *Lyrodus affinis* | PCMB51881, MAL032295 | Honolulu H, Oʻahu, HI, USA; TP, 2 m; 21.303865°, −157.870435° | this study | LREG013−23 | X | X | | |
| *Lyrodus affinis* | PCMB51882, MAL032296 | Honolulu H, Oʻahu, HI, USA; TP, 2 m; 21.303865°, −157.870435° | this study | LREG014−23 | X | | | |
| *Lyrodus affinis* | PCMB51885, MAL032299 | Waiʻanae H, Oʻahu, HI, USA; TP, 1 m; 21.449683°, −158.197700° | this study | LREG015−23 | X | | | X |
| *Lyrodus affinis* | PCMB54284, MAL010723 | Makai Pier, Oʻahu, HI, USA; TP, 1 m; 21.317881, −157.669195⁰ | this study | LREG018−23 | X | X | | |
| *Lyrodus affinis* | PCMB54294, MAL010733 | Makapuʻu Bh, Oʻahu, HI, USA; Df, 0.5 m; 21.296236°, −157.658150° | this study | LREG020−23 | X | | | |
| *Lyrodus affinis* | PCMB54296, MAL010735 | James Campbell Nat. W. Refuge, Oʻahu, HI, USA; Df, itt; 21.702004°, −157.958956° | this study | LREG021−23 | X | | | |
| *Lyrodus affinis* | PCMB54300, MAL010739 | Makapuʻu Bh, Oʻahu, HI, USA; Df, 0.5 m; 21.296236°, −157.658150° | this study | LREG022−23 | X | | | |
| *Lyrodus affinis* | PCMB55052, MAL010768 | Waiʻanae H, Oʻahu, HI, USA; TP, 1 m; 21.449683°, −158.197700° | this study | LREG025−23 | X | | | |
| *Lyrodus affinis* | PCMB55889, MAL010779 | Hanamāʻulu Bay, Kauaʻi, HI, USA; Df, 0.5 m; 21.994851°, −159.341029° | this study | LREG026−23 | X | | | |
| *Lyrodus affinis* | PCMB55987, MAL010784 | Kahana Bay, Oʻahu, HI, USA; Df, 0.5 m; 21.558569°, −157.866637° | this study | LREG029−23 | X | | | |
| *Lyrodus* cf. *floridanus* | PCMB50795, MAL032270 | Charlie Pier, Pearl H, Oʻahu, HI, USA; TP, 4 m; 21.351456°, −157.964805° | this study | LREG001−23 | X | X | X | X |
| *Lyrodus* cf. *floridanus* | PCMB50798, MAL302273 | Menehune Fish Pond, Nāwiliwili E, Kauaʻi, HI, USA; Df, itt; 21.947965°, −159.372238° | this study | LREG002−23 | X | X | X | |
| *Lyrodus* cf. *floridanus* | PCMB50801, MAL032276 | Nāwiliwili H, Kauaʻi, HI, USA; Df, 1 m; 21.949860°, −159.357818° | this study | LREG003−23 | X | X | | |
| *Lyrodus* cf. *floridanus* | PCMB51879, MAL032293 | Heʻeia, Oʻahu, HI, USA; Df, itt; 21.440051°, −157.809488° | this study | LREG012−23 | X | | X | |
| *Lyrodus* cf. *floridanus* | PCMB56992, MAL032327 | Coconut Is, Oʻahu, HI, USA; TP, 1 m; 21.432773°, −157.788809° | this study | LREG030−23 | X | | | |
| *Lyrodus* cf. *floridanus* | PCMB67921, MAL032262 | Kapapapuhi Park, Pearl H, Oʻahu, HI, USA; Df, itt; 21.366573°, −158.016850° | this study | LREG033−23 | X | X | | |
| *Lyrodus* cf. *floridanus* | PCMB67922, MAL032264 | Coconut Is, Oʻahu, HI, USA; TP, 1 m; 21.432773°, −157.788809° | this study | LREG034−23 | X | X | | |
| *Lyrodus* cf. *floridanus* | PCMB61141 | Hilo H, Hawaiʻi Is., USA; TP, 1m; 19.731261°, −157.964820° | this study | LREG032−23 | X | | | |
| *Lyrodus medilobatus* | PCMB51889, MAL032303 | Waiʻanae H, Oʻahu, HI, USA; TP, 1 m; 21.449683°, −158.197700° | this study | LREG017−23 | X | X | X | X |
| *Lyrodus medilobatus* | PCMB54288, MAL010727 | Waiʻanae H, Oʻahu, HI, USA; TP, 1 m; 21.449683°, −158.197700° | this study | LREG019−23 | X | X | | |

*(Continued)*

| ID | Specimen | Locality | Reference | BOLD or GenBank Numbers | COI | 16S | 18S | 28S |
|---|---|---|---|---|---|---|---|---|
| *Lyrodus medilobatus* | PCMB55033, MAL010749 | Waiʻanae H, Oʻahu, HI, USA; TP, 1 m; 21.449683°, −158.197700° | this study | LREG023−23 | X | | | |
| *Lyrodus medilobatus* | PCMB55891, MAL010781 | Anahola, Kauaʻi, HI, USA; Df, 1m; 22.156766°, −159.305519° | this study | LREG027−23 | X | | | |
| *Lyrodus medilobatus* | PCMB55892, MAL010782 | Anahola, Kauaʻi, HI, USA; Df, 0.5 m; 22.156766°, −159.305519° | this study | LREG028−23 | X | | | |
| *Lyrodus medilobatus* | PCMB56996, MAL032332 | Tug Pier, Sand Island, Midway Atoll; TP, 1 m; 28.214652°, −177.363430° | this study | LREG031−23 | X | | | |
| *Lyrodus* cf. *mersinensis* | PCMB55051, MAL010767 | Tomioka Bay, Amakusa, Kumamoto Pref., Japan; TP 2 m; 32.527463°, 130.034576° | this study | LREG024−23 | X | X | X | X |
| *Lyrodus mersinensis* | WBET134 | Erdemli/Mersin, Turkey; TP, 2 m; 36.5653⁰, 34.2562⁰ | [10] | KC157939; KC158218 | X | | X | |
| *Lyrodus mersinensis* | WBET130 | Erdemli/Mersin, Turkey; TP, 2 m; 36.5653⁰, 34.2562⁰ | [10] | KC157938; KC158217 | X | | X | |
| *Lyrodus mersinensis* | WBET133 | Erdemli/Mersin, Turkey; TP, 2 m; 36.5653⁰, 34.2562⁰ | [10] | KC157916 | X | | | |
| *Lyrodus mersinensis* | WBET135 | Erdemli/Mersin, Turkey; TP, 2 m; 36.5653⁰, 34.2562⁰ | [10] | KC157932; KC158211 | X | | X | |
| *Lyrodus* cf. *pedicellatus* | PCMB51851, MAL032307 | Pt. Cavallo, San Francisco Bay, CA, USA; Df, itt; 37.832244°, −122.474830° | this study | LREG006–23 | X | X | X | X |
| *Lyrodus* cf. *pedicellatus* | PCMB51852, MAL032308 | Pt. Cavallo, San Francisco Bay, CA, USA; Df, itt; 37.832244°, −122.474830° | this study | LREG007–23 | X | X | | |
| *Lyrodus pedicellatus* | PCMB51853, MAL032309 | Pt. Cavallo, San Francisco Bay, CA, USA; Df, itt; 37.832244°, −122.474830° | this study | LREG008–23 | X | X | X | X |
| *Lyrodus pedicellatus* | WBET078 | Golfe du Morbihan, Brittany, France; TP, 2 m; 47.590000°, −2.790000° | [10] | KC157915; KC158196 | X | | X | |
| *Lyrodus pedicellatus* | WBET118 | Toulindac, Brittany, France; TP, 2 m; 47.5977, −2.8616 | [10] | KC157917 | X | | | |
| *Lyrodus pedicellatus* | WBET087 | Toulindac, Brittany, France; TP, 2 m; 47.5977, −2.8616 | [10] | KC157919 | X | | | |
| *Lyrodus pedicellatus* | WBET090 | Toulindac, Brittany, France; TP, 2 m; 47.5977, −2.8616 | [10] | KC157920; KC158201 | X | | X | |
| *Lyrodus pedicellatus* | WBET042 | Golfe du Morbihan, Brittany, France; TP, 2m; 47.590000°, −2.790000° | [10] | KC157921 | X | | | |
| *Lyrodus pedicellatus* | WBET043 | Golfe du Morbihan, Brittany, France; 47.590000°, −2.790000° | [10] | KC157922; KC158203 | X | | X | |
| *Lyrodus pedicellatus* | WBET061 | Beder, Brittany, France; 47.580000°, −2.880000° | [10] | KC157937; KC158216 | X | | X | |
| *Lyrodus pedicellatus* | BMNH 20070253 | Portsmouth, UK | [49] | AM774540; AM779714 | | | X | X |
| *Lyrodus pedicellatus* | Ro275 | Atlantic Ocean | [34] | KU201129 | | | | X |
| *Lyrodus pedicellatus* | Ro258 | Mediterranean | [34] | KU201127 | | | | X |
| *Lyrodus pedicellatus* | OGLS00502 | Florida, USA | [34] | JF899211, JF899184 | | | X | X |
| *Lyrodus pedicellatus* | Ro256 | Atlantic Ocean | [34] | KU201120 | | | X | |
| *Lyrodus pedicellatus* | Ro274 | Atlantic Ocean | [34] | KU201121 | | | X | |
| *Lyrodus reginae* sp. nov. | PCMB51886, MAL032300 | Waiʻanae Harbor, Oʻahu, HI, USA; TP, 1 m; 21.449683°, −158.197700° | this study | LREG016–23 | X | X | | |
| *Lyrodus reginae* sp. nov. | PCMB51877, MAL032291 | Charlie Pier, Pearl H, Oʻahu, HI, USA 21.351456°, −157.964805° | this study | LREG010–23 | X | X | X | X |

*(Continued)*

**Table 2.** (Continued)

| ID | Specimen | Locality | Reference | BOLD or GenBank Numbers | COI | 16S | 18S | 28S |
|---|---|---|---|---|---|---|---|---|
| *Lyrodus reginae* sp. nov. | PCMB51878, MAL032292 | Charlie Pier, Pearl H, Oʻahu, HI, USA, 21.351456°, −157.964805° | this study | LREG011–23 | X | X | | |
| *Lyrodus* sp. | G0 FL-09 | Merrit Is, Florida; 28.406000°, −80.660300° | [50] | OM910820.1 | X | X | | |
| *Lyrodus takanoshimensis* | PCMB51854, MAL032310 | Tomioka Bay, Amakusa, Kumamoto Pref., Japan; TP, 2 m; 32.527463°, 130.034576° | this study | LREG009–23 | X | X | X | X |
| *Lyrodus takanoshimensis* | OGL2 | | OGL | KY250361 | | | X | |

Bh: beach; Df: driftwood; E: estuary; H: harbor; itt: intertidal; m: depth in meters; MAL: BPBM Malacology; OGL: Ocean Genomic Legacy; PCMB: Pacific Center for Molecular Biodiversity; Pref: prefecture; TP: test panel.

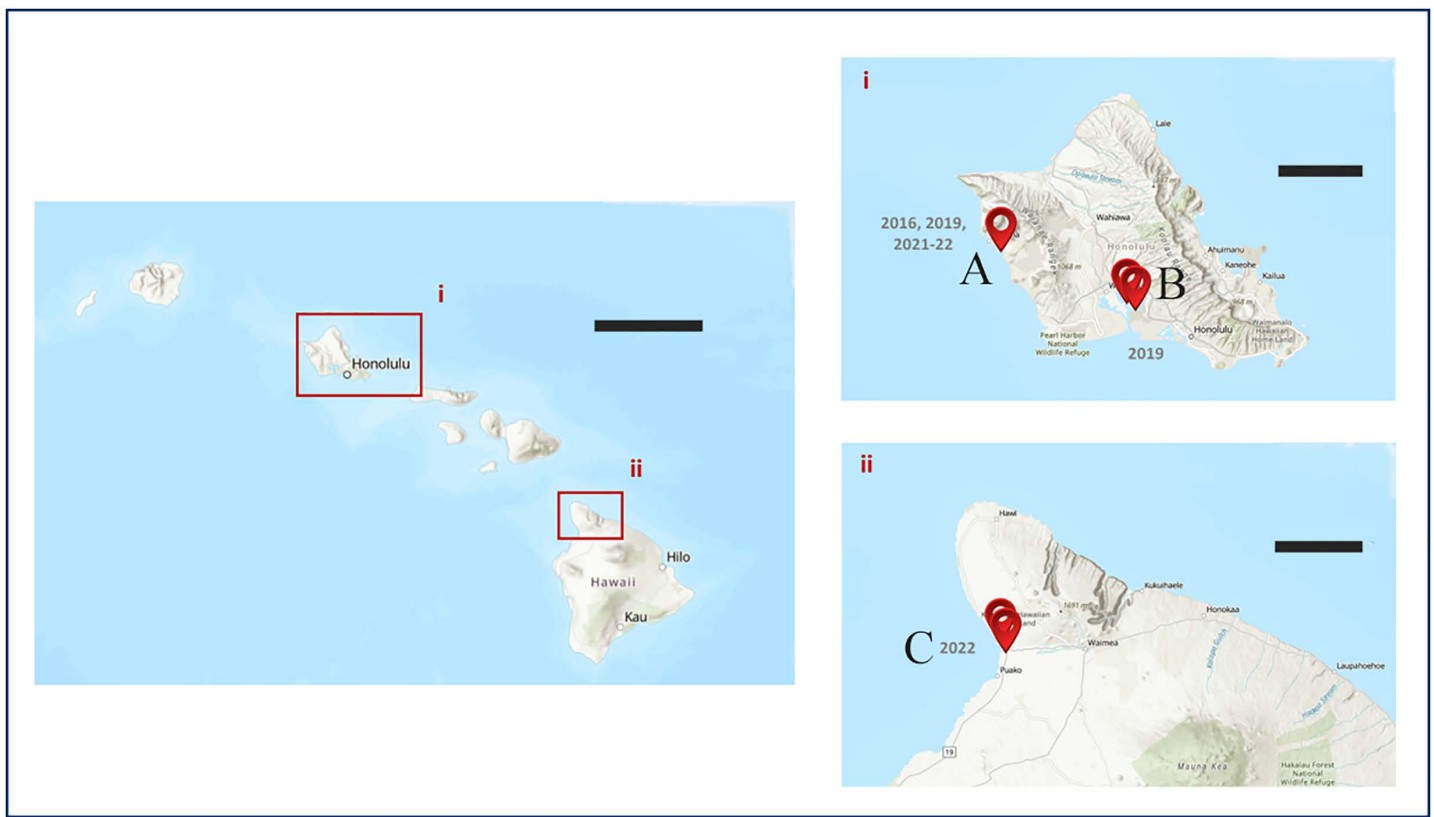

**Fig 2. *Lyrodus reginae* sp. nov. collection locations and dates.** Left: an overview of Hawaiʻi, scale = 100 km. (i) Oahu, scale = 16 km. A: Waiʻanae 2016, 2019, 2021, 2022; B: Pearl Harbor, Charlie Pier 2019 and Ford Island 2022. (ii) Northwest Hawaiʻi Island, scale = 16 km. C: Pelekane estuary and Spencer Beach Park 2022. Map created using USGS map reader.

Initially identified as *L. pedicellatus*, this SqLTB was later determined to be an undescribed species, *L. reginae* sp. nov. described herein (Figs 3, 4A, 4B and 5A).

The remaining 114 *Lyrodus pedicellatus*-like brooding specimens were STBs, with D-stage or younger larvae in the gills, and gill color ranging from white to lavender. We assign the name *L.* cf. *floridanus* to this STB until further analysis

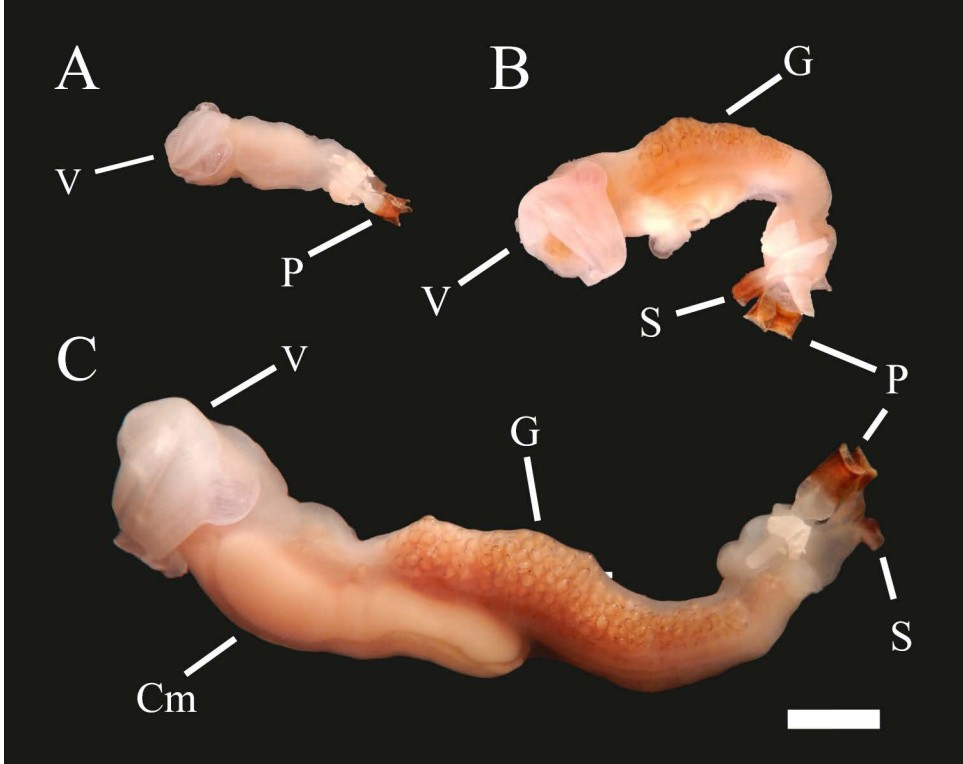

**Fig 3. *Lyrodus reginae* sp. nov. (A)** Juvenile specimen (PHCO19A-12, NCT PC). **(B)** Paratype, small adult, brooding pediveliger larvae in the gills (BPBM 286280). **(C)** Holotype, fully mature adult, brooding pediveliger larvae in the gills (BPBM 286279). Cm, cecum; G, gill; P, pallet; S, siphon; V, valves. Scale bar = 2 mm.

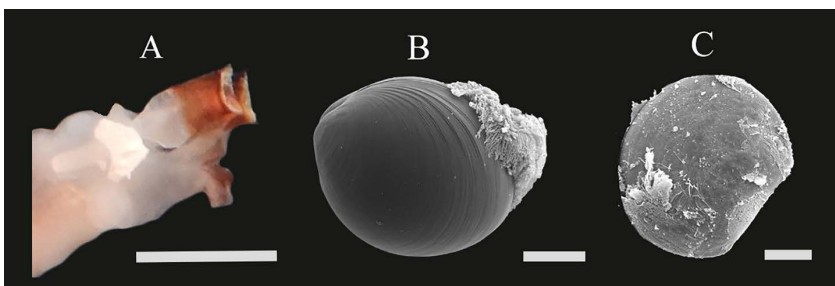

**Fig 4. Posterior end of *L. reginae* and larvae of *L. reginae* and *L.* cf. *floridanus*. (A)** Posterior end of *Lyrodus reginae* sp. nov. Holotype, BPBM 286279, scale bar = 2 mm. **(B)** Pediveliger larvae of *L. reginae* sp. nov., WaHODF16−5, NCT PC, scale bar = 100 μm. **(C)** D-stage larvae of *L.* cf. *floridanus*, PHWODF22−10, JRS PC, scale bar = 20 μm.

and sampling can be used to clarify its identity (Figs 4C, 5B and 6). This species is abundant on Oʻahu, Kauaʻi, and Hawaiʻi Island (20, 6, and 4 sites, respectively), most often in harbors, bays, mangrove lagoons, estuaries, or locations protected from direct wave action by reefs and/or lava barriers. Molecular delimitation shows this is a separate species from the STBs *L. mersinensis* and *L.* cf. *mersinensis* (Fig 7).

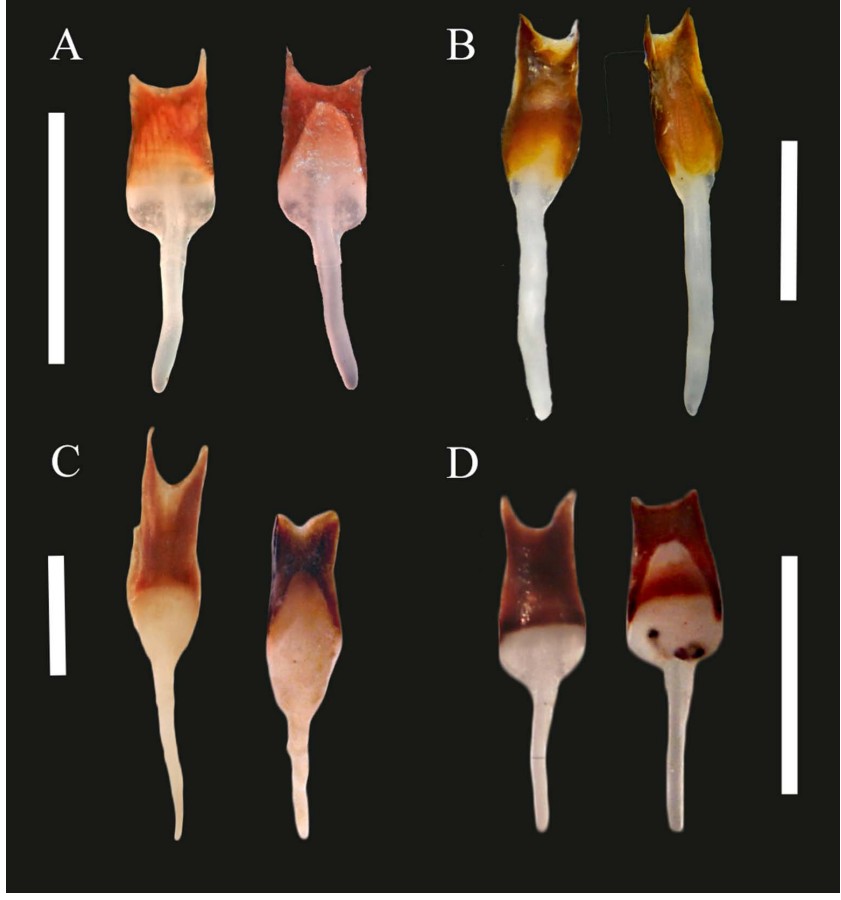

**Fig 5. Pallets of *Lyrodus pedicellatus* species complex.** Left: outer face, right: inner face. **(A)** *L. reginae,* sp. nov. (Oʻahu, HI, USA; BPBM 293709). **(B)** *L.* cf. *floridanus* (Oʻahu, HI, USA; MCZ:Mala:420735). **(C)** *L. mersinensis* (Mersin, Turkey, PC LMSB). **(D)** *L. pedicellatus* (Terceira, Azores, Portugal, PC LMSB). Scale bars = 2 mm.

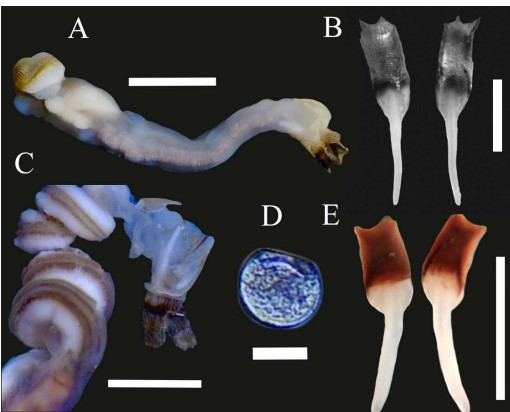

**Fig 6. *Lyrodus* cf. *floridanus*. (A)** Whole animal with D stage larvae, note lavender color of gills, HHOOK17−2, NCT PC, scale bar = 4 mm. **(B)** Pallet, left: outer face, right inner face, BPBM 288830, scale bar = 2 mm. **(C)** Posterior end with gills at early 'white' stage of brooding, BPBM 288830, scale bar = 4 mm. **(D)** Straight-hinge D stage larvae, BPBM 285340, scale bar = 48 μm. **(E)** Pallet, left: outer face, right: inner face, HHBOK22b-4, NCT PC, scale bar = 2 mm.

**Fig 7. Bayesian inference phylogenetic reconstruction of shipworm relationships for *Lyrodus* species with a focus on the *Lyrodus pedicellatus* complex.** Node values represent posterior probabilities from 8 million generations of 4 chain analysis, and shaded circles for values from 20,000 Ultrafast Bootstrap replicates under a maximum likelihood estimation in IQTree. Colored clades/partitions correspond to a prior identified species via pallet and reproductive morphology and those delineated using ASAP and bPTP. Geographic locations for collections of individuals in each clade are provided.

Six specimens with SqLTB *L. pedicellatus* morphology were collected from one of the six pieces of submerged driftwood collected in San Francisco Bay. Three of these specimens were sequenced. Our phylogenetic analysis found one to be *L. pedicellatus*, the other two were of an undescribed species, referred to herein as *L.* cf. *pedicellatus* (Fig 7).

One test panel from Amakusa contained a STB species with *L. pedicellatus* pallet morphology. We sequenced one specimen, which was determined to be a closely related, but a new and separate species from *L. mersinensis*, which we designate here as *L.* cf. *mersinensis* (Fig 7).

## Larval brooding strategy

*Lyrodus reginae* sp. nov. is a long-term sequential brooder (SqLTB) with gills containing larvae in an age class series from embryo to pediveliger larvae (Figs 3B, 3C and 4B). Specimens were confirmed as sexually mature at small sizes (~4 mm; Fig 3B) based on the presence of brooded larvae. (Figs 3B, 3C and 4B). In addition, two other SqLTB species were collected, *L. pedicellatus* and *L.* cf. *pedicellatus*.

*Lyrodus* cf. *floridanus* was confirmed as a short-term brooder (STB), spawning larvae at the straight-hinge, D stage (Figs 4C and 6D). The gills become bright white as the larvae mature (Fig 6C), and at the D stage take on a lavender color due to the reflection of light off the larval shells (Fig 6A). Brooding was observed in specimens with body size as small as 32 mm, with 61% of brooding individuals at the early, white gill state, and 39% with lavender gill color. Examination of the holotype of *L. mersinensis* revealed it is a STB, as is the undescribed species from Japan, *L. cf. mersinensis*.

## Comparative material examined

The neotype of *Lyrodus pedicellatus* (MNHN-IM-2000–32924) is a body in two parts with larvae from embryo to pediveliger in the gills (SqLTB). The holotype of *L. mersinensis* (MNHN-IM-2000–33821) consists of the posterior end with intact siphons and pallets only and does not feature the anterior region including the valves. An examination of a *L. mersinensis* paratype (MNHN-IM-2000–33822) found the gills packed with straight-hinge D larvae, loose in the gill cavity, with the typical lavender color of a late-stage STB. Turner [1, pg. 132] reported the single pallet of the *L. floridanus* syntype (USNM 193031) lost, with only shells remaining. NCT examined the syntype in May 2023, finding only shells, confirming the loss of the pallet.

In addition to the types of currently accepted species within the *Lyrodus pedicellatus* complex, type material of several historical species synonymized by Turner [1] with *L. pedicellatus* were examined. The syntype of *Teredo kauaiensis* Dall, Bartsch, & Rehder, 1938 (USNM 508681, 11 pallets, a mix from Honolulu Harbor, Oʻahu and Nāwiliwili Harbor, Kauaʻi), and the holotypes of *Teredo diegensis* Bartsch, 1916 (USNM 74219), *Teredo (Teredops) diegensis var. midwayensis* Edmondson, 1946 (BPBM 9957), *Teredo samoaensis* R. C. Miller, 1924 (CAS 066381), and *Teredo yatsui* Moll, 1929 (ZMB 108.897) all had pallets with typical *L. pedicellatus* morphology. The two pallets of the *Teredo honoluluensis* Edmondson, 1946 holotype (BPBM 9955) are flattened at the distal end of the blade, a shape inconsistent with the cone-shaped blade of *L. pedicellatus*, and the majority of the periostracal caps are gone.

Our examination of shipworm specimens (BPBM Malacology) collected by Edmondson discovered SqLTB individuals with *Lyrodus pedicellatus* pallet morphology from Oʻahu and Midway Atoll. Edmondson designated these as *L. diegensis* Bartsch, 1916 (synonymized by Turner [1] with *L. pedicellatus*). The number of bodies in these lots greatly exceeds those collected during our study; for example, BPBM 289049 from Honolulu Harbor, Oʻahu in 1941, contained 154 individuals, all brooding pediveliger larvae. A STB with *L. pedicellatus* pallet morphology was collected in 1941 by Edmondson at Hilo, Hawaiʻi Island (collection representatives: BPBM 293746, 293732). Two lots from Florida deposited by Calloway at the MCZ, identified as *L. floridanus* by Calloway, either lacked pallets (MCZ:Mala:350276) or were not brooding (MCZ:Mala:356853). Consequently, confirmation of these specimens' identity was not possible. Five specimens, collected in 1978 on Oʻahu (MCZ:Mala:350275), were STBs of the *L. pedicellatus* complex. We designate the SqLTB from Edmondson's surveys as '*L. pedicellatus* clade' due to the lack of molecular confirmation of species identity and the uncertainty of the number of cryptic species in this complex, and the STB in Hawaiian waters as *L.* cf. *floridanus*.

## Molecular results

For this study, 37 new *COI* sequences representing nine shipworm taxa (Table 2), including three specimens from *Lyrodus reginae* sp. nov., were generated and combined with 12 sequences from GenBank to produce a *COI* alignment of 1,145 bp. The newly generated sequences encompass and extend previously reported variation.

The new *COI* sequences are longer and approximately 250 bp in from the 5' end of the standard 650 bp barcoding fragment. The other three loci produced alignments consisting of 523 bp (*16S*), 1,801 bp (*18S*), and 1,499 bp (*28S*). The final concatenated matrix with all four loci included 52 OTUs with 4,961 bp, including gaps and missing data.

The mean pairwise K2P genetic distances at *COI* among *Lyrodus* species ranged from a low of 0.154 (± 0.015 SE) between *L. pedicellatus* and *L.* sp. from Florida, to a high of 0.234 (± 0.018 SE) between *L. affinis* and *L. takanoshimensis*. The *L. pedicellatus* complex had a pairwise mean K2P distance of 0.15 (± 0.010 SE), ranging from 0.154 (± 0.015 SE)

between *L. pedicellatus* and *L.* sp. from Florida, and 0.227 (± 0.021 SE) between *Lyrodus* sp. and *L. mersinensis*. The pairwise K2P distances between *Lyrodus reginae* sp. nov. and *Lyrodus pedicellatus*, with which it had previously been confused, was 0.163 (± 0.018 SE). Individuals belonging to the two clades of *Lyrodus mersinensis* differed by 0.025 (± 0.008 SE), and those clustering in two clades of *L. pedicellatus* differed by 0.035 (± 0.008 SE).

DeSigNate identified 50 distinct nucleotide states among the *COI* sequences of *L. reginae* sp. nov. when compared with the other taxa in the *Lyrodus pedicellatus* complex (Table 3). The other three loci contained 17 (*16S*), 5 (*18S*), and 8 (*28S*) nucleotide synapomorphies (Tables 4–6, respectively) or molecular diagnostic characters, distinguishing *L. reginae* sp. nov. from other taxa in the complex.

Phylogenetic reconstruction under ML and BI produced identical topologies with overall similar levels of support at all nodes (Fig 7). Although pairwise *COI* divergence between the major clades reached 0.25, we do not interpret this value as a threshold or criterion for species designation. Rather, we note that divergence levels of this magnitude are typically observed among recognized species rather than within them across numerous molluscan lineages, and when considered together with independent evidence from phylogenetic topology, geography, and brooding mode, support recognition of these groups as distinct species.

The *Lyrodus pedicellatus* complex was well supported, with *Lyrodus reginae* sp. nov. sister to the clade containing *L. mersinensis, L.* cf. *mersinensis* and *L.* cf. *floridanus*. This clade was in turn sister to the clade comprised of *L. pedicellatus* and an undescribed *Lyrodus* species from Florida. The remaining three species of *Lyrodus* were recovered in a weakly supported clade with an unresolved relationship to the *Lyrodus pedicellatus* complex. The number of species defined by both delimitation approaches were largely congruent with each other and with node support in the inferred phylogeny. The rank-1 ASAP partition corresponded to ten species-level clades recovered in phylogenetic analyses, whereas bPTP supported eleven putative species due to the subdivision of one phylogenetic lineage with limited support from additional molecular or biological evidence (Fig 7).

Our phylogenetic analysis recovered seven genetically distinct lineages within the *Lyrodus pedicellatus* morphospecies complex, three SqLTBs: *L. reginae* sp. nov. (Hawaiʻi), *L. pedicellatus* (France, San Francisco), and *L.* cf. *pedicellatus* (San Francisco), and three STBs: *L.* cf. *floridanus* (Hawaiʻi), *L. mersinensis* (Turkey), and *L.* cf. *mersinensis* (Japan). Brooding mode remains undetermined in the 7th species, *Lyrodus* sp. (Florida). In the case of *L.* cf. *pedicellatus* and *L.* cf. *mersinensis*, only one specimen of each was sequenced, and only the former was delimitated by both software programs. All characters (morphological and molecular) may be modified with future sampling, aligning with best practices in integrative taxonomy.

## Systematics

**Teredinidae Rafinesque, 1815**
***Lyrodus* Gould, 1870**
**Type species*: Lyrodus pedicellatus* (Quatrefages, 1849)**
***Lyrodus reginae* Treneman, DeLeon, Shipway, Borges & Hayes sp. nov.**
(Figs 3C and 4A)
**Zoobank ID**: urn:lsid:zoobank.org:act:70EA3556-C653-4C43-9DD7-CEBE69ACB6C1
***Synonymy***: None, see remarks.
*Holotype*: BPBM 286279, whole body in two parts, measuring 17 mm in total body length (Figs 3C and 4A). Long-term, sequential brooder featuring pediveliger larvae in the gills. Collected by NCT from a Douglas fir panel, deployed August 8th to December 17th, 2019 at a depth of 1 meter in Waiʻanae Harbor, Oʻahu, Hawaiʻi, USA, and preserved in 95% ethanol. Paratype: 286280 (Fig 3B).
***Type Locality***: Waiʻanae Harbor, Oʻahu, Hawaiʻi, USA (21.44940, −158.19750) (Fig 2i).

**Etymology**. *reginae* (noun) in honor of Regina Kawamoto, marine mollusc collections technician at the Bernice P. Bishop Museum and Research Associate at the University of Hawaiʻi at Mānoa, for her dedication to Malacology.

**Table 3.** *COI* Synapomorphies: Molecular diagnostic characters (MDCs) derived from *COI* sequences for species within the *Lyrodus pedicellatus* complex.

| Alignment Position | L. reginae sp. nov. (3) | L. cf. mersinensis (1) | L. mersinensis (4) | L. cf. floridanus (8) | Lyrodus sp. (1) | L. cf. pedicellatus (2) | L. pedicellatus (8) |
|---|---|---|---|---|---|---|---|
| 296 | T | G | G | G | A | G | A |
| 299 | T | A | A | A | A | A | A |
| 305 | C | T | T | T | T | T | T |
| 306 | T | C | C | C | C | C | C |
| 323 | C | T | T | T | T | T | T |
| 338 | G | T | T | T | A | T | A |
| 344 | C | G | G | A/T | G | G | G |
| 374 | G | T | T | A | T | T | T |
| 398 | G | T | T | T | T | T | T |
| 404 | C | A | A | T | T | T | T |
| 431 | T | G | G | A | G | A | A |
| 443 | G | T | T | T | T | T | T |
| 446 | A | T | T | C | T | T | T |
| 458 | A | T | T | T | T | T | T |
| 468 | T | C | C | C | C | C | C |
| 470 | A | T | T | T/C | G | T | C |
| 482 | G | A | A | A | A | A | A |
| 501 | A | G | G | G | G | G | G |
| 521 | G | A | A | A | T | T | T |
| 584 | G | A | A | T | A | A | A |
| 644 | G | T | T | T | A | T | T |
| 698 | A | T | ? | T | T | T | T |
| 716 | C | T | ? | T | T | T | T |
| 717 | T | C | ? | C | C | C | C |
| 719 | A | T | ? | T | T | T | T |
| 731 | T | G/? | ? | G | G | G | A/? |
| 752 | G | A | ? | A | A | A | A |
| 758 | T | G/? | ? | G | C | C | C/? |
| 782 | A | G? | ? | T | G | T | T/? |
| 785 | G | A | ? | A | A | A | A |
| 818 | G | A/? | ? | T | A | A | A/? |
| 821 | T | G/? | ? | A | G | A | A/? |
| 842 | G | T/? | ? | T | T | C | T? |
| 848 | C | T | ? | T | T | T | T |
| 881 | G | T | ? | T | T | T | T |
| 887 | C | T | ? | T | T | T | T |
| 899 | G | A/? | ? | T | T | T | T/? |
| 908 | C | T | ? | T | T | T | T |
| 920 | A | T | ? | T | T | T | T |
| 923 | G | T | ? | T | T | T | T |
| 926 | T | A/? | ? | G | A | A | A/? |
| 965 | A | T/? | ? | C | G | T | T/? |
| 983 | G | T/? | ? | T | A | A | A/? |
| 1013 | T | G/? | ? | A | A | G | G/? |

*(Continued)*

**Table 3.** (Continued)

| Alignment Position | L. reginae sp. nov. (3) | L. cf. mersinensis (1) | L. mersinensis (4) | L. cf. floridanus (8) | Lyrodus sp. (1) | L. cf. pedicellatus (2) | L. pedicellatus (8) |
|---|---|---|---|---|---|---|---|
| 1022 | T | G/? | ? | G | A | A | A/? |
| 1031 | A | G | ? | G | G | G | G |
| 1037 | A | T | ? | T | T | T | T |
| 1074 | T | ? | ? | C | C | C | C |
| 1076 | A | ? | ? | T | T | T | T |
| 1094 | G | T/? | ? | T | A | A | A/? |

Diagnostic characters represent fixed nucleotide states observed in the sampled material. Alignment positions refer to the *COI* alignment used for phylogenetic analyses. The number of sequences analyzed per species is indicated in parentheses in the column headers. Question marks designate missing data from shorter sequences; base call with a question mark indicates one sequence from this study was in the group with GenBank sequences that contained missing data.

**Table 4.** *16S* molecular synapomorphies for *L. pedicellatus* complex taxa.

| Alignment Position | L. reginae sp. nov. (3) | L. cf. mersinensis (1) | L. cf. floridanus (5) | Lyrodus sp. (1) | L. cf. pedicellatus (2) | L. pedicellatus (1) |
|---|---|---|---|---|---|---|
| 27 | C | T | T | T | T | T |
| 31 | A | G | G/C | T | C | T |
| 46 | G | A | A | A | A | A |
| 167 | A | T | T | T | G | G |
| 178 | C | T | T | T | T | T |
| 183 | G | A | A | A | T | T |
| 184 | G | A | A | A | T | T |
| 220 | T | A | G | A | C | C |
| 221 | C | G | G | A | T | T |
| 267 | C | A | T | T | T | T |
| 275 | G | C | C | T | T | T |
| 288 | G | A | A | A | A | A |
| 309 | G | C | T | A | A | A |
| 329 | A | T | T | T | T | T |
| 330 | G | A | T | A | T | T |
| 337 | G | A | A | T | A | A |
| 465 | T | C | C | C | C | C |

Number of sequences for each species is in parentheses.

**Table 5.** *18S* molecular synapomorphies for *L. pedicellatus* complex taxa.

| Alignment Position | L. reginae sp. nov. (1) | L. cf. mersinensis (1) | L. cf. floridanus (3) | Lyrodus sp. (0) | L. cf. pedicellatus (1) | L. pedicellatus (9) |
|---|---|---|---|---|---|---|
| 646 | T | C | C | – | C | C |
| 783 | C | T | T | – | T | T |
| 1040 | T | A | A | – | A | A |
| 1394 | T | G | G | – | G | G |
| 1742 | T | ? | C | – | C | ? |

Number of sequences for each species is in parentheses. Base call with a question mark indicates one sequence from this study was in the group with GenBank sequences that contained missing data. - indicates taxon had no sequence for this locus.

**Table 6.** *28S* molecular synapomorphies for the *L. pedicellatus* complex taxa.

| Alignment Position | *L. reginae* sp. nov. (1) | *L.* cf. *mersinensis* (1) | *L.* cf. *floridanus* (1) | *Lyrodus* sp. (0) | *L.* cf. *pedicellatus* (1) | *L. pedicellatus* (5) |
|---|---|---|---|---|---|---|
| 41 | C | ? | T | – | T | ? |
| 264 | C | T | T | – | T | ? |
| 432 | T | C | C | – | C | ? |
| 437 | C | T | T | – | T | ? |
| 526 | T | C | C | – | C | ? |
| 763 | T | C | C | – | C | ? |
| 1195 | C | ? | ? | – | ? | ? |
| 1196 | A | ? | ? | – | ? | ? |

Number of sequences for each species is in parentheses. – indicates no sequences available for those taxa. Base call with a question mark indicates one sequence from this study was in the group with GenBank sequences that contained missing data.

**Table 7.** Comparison of four of the seven cryptic species within the *Lyrodus pedicellatus* complex.

| | *L. reginae* sp. nov. | *L. pedicellatus* | *L.* cf. *mersinensis* | *L.* cf. *floridanus* |
|---|---|---|---|---|
| **Distribution** | Hawai'i: O'ahu, Hawai'i Is. | North East Atlantic, San Francisco Bay | Japan | Hawai'i: O'ahu, Kaua'i, Hawai'i Is. |
| **Habitat** | Hawai'i: island west sides, Pearl Harbor | bays, estuaries, open coast | bay | bays, harbors, estuaries, protected shores |
| **Larval brooding strategy** | SqLTB | SqLTB | STB | STB |
| **Reference** | this study | [10,56], this study | [57], this study | this study |

SqLTB: sequential long-term brooder; STB: short-term brooder.

**Diagnosis**. Pallet morphology and non-brooding gross tissue anatomy as in *Lyrodus pedicellatus*, according to Turner [1,55]. Gross tissue anatomy of brooding specimens as in sequential long-term brooders, such as *L. pedicellatus*, but unlike *L. mersinensis* and *L. floridanus* which are short-term brooders (Table 7 and Figs 3, 4A and 5), according to Calloway and Turner [15].

**Molecular diagnosis**. Molecular differentiation among species in the *Lyrodus pedicellatus* complex is supported by fixed nucleotide character states across multiple loci. *COI* sequences exhibit 50 molecular diagnostic characters (MDCs) distributed between alignment positions 296 and 1094 bp across the seven species examined (Table 3). The *16S rRNA* gene contains 17 MDCs spanning positions 27–465 bp (Table 4), while the *18S rRNA* gene shows five MDCs between positions 646 and 1742 bp (Table 5). The *28S rRNA* gene includes eight MDCs located between positions 41 and 1196 bp (Table 6).

**Comparative material**. *Lyrodus pedicellatus*, neotype, MNHN-IM-2000–32924; *Lyrodus mersinensis*, holotype, MNHN-2000–33821; *L. mersinensis*, paratypes: MNHN-IM-2000–33822, −22823, −33824; *L. floridanus*, syntype, USNM 193031; *Teredo diegensis*, holotype, USNM 74219; *T. (Teredops) diegensis var. midwayensis,* holotype BPBM 9957*; T. kauaiensis*, syntype, USNM 508681; *T. samoaensis*, holotype, CAS 066381; *T. yatsui,* holotype, ZMB 108.897.

**Habitat**. Submerged wood in near-shore waters of western and southern O'ahu and northwest Hawai'i Island. Found at salinities 31–34 PSU and temperatures 24–29°C. Wood panels submerged during summer (June to September) were found to more commonly contain *L. reginae* sp. nov.

**Distribution.** Currently known only in Hawai'i: Pearl Harbor, O'ahu, and the west sides of O'ahu and Hawai'i Island (Fig 2).

**Description**. Pallet morphology and non-brooding gross tissue anatomy indistinguishable from other species in the *L. pedicellatus* complex: *L. pedicellatus, L. floridanus, L. mersinensis, L.* cf. *mersinensis, L.* cf. *floridanus* (Hawai'i), *L.* cf

*pedicellatus* (San Francisco), *Lyrodus* sp. (Florida) (Table 7 and Figs 3, 4A, 5 and 6). Small, observed body length range: 3–40 mm (n = 12). Pallets (length range 1.5–3 mm) consist of a calcareous stalk and distal calcareous oval to cone-shaped blade, at times with a slightly convex outer surface and slightly concave inner surface. Smooth blade and stalk with no obvious layering, often slightly transparent. Blade with distal light to dark brown, sometimes semi-transparent periostracal cap. Cap covering and extending beyond distal half of blade, with shallow U-shaped distal edge, sometimes with lateral horns. Sequential long-term brooder. Individuals reach maturity and begin brooding at small size (4 mm). Gills of brooding individuals packed with golden-brown pediveliger larvae proximally, followed distally by a gradient of younger and younger larvae. Tunnel lengths ranged from 8–35 mm.

**Remarks**. *Lyrodus pedicellatus* (type locality: Pasajes Port, San Sebastian, Spain) was viewed as a cosmopolitan morphospecies, after Turner [1,55] synonymized 32 species described from around the world into this one species. When both STB and SqLTB populations were discovered in this morphospecies Calloway and Turner [16] resurrected *L. floridanus* as the species designation for the STB.

The type localities for several *L. pedicellatus* synonyms are within the Hawaiian Archipelago: *Teredo hawaiensis* Dall, Bartsch & Rehder, 1938 (offshore, Oʻahu), *T. honoluluensis* Edmondson, 1946 (Honolulu Harbor, Oʻahu), *T. kauaiensis* (Nāwiliwili, Kauaʻi), and *T. diegensis* var *midwayensis* Edmondson, 1946 (Midway Atoll). The type material and descriptions of these synonyms lack information on their reproductive modes, have pallets resembling the *L. pedicellatus* morphotype, and/or are missing pallet descriptions (*Teredo hawaiensis* Dall, Bartsch, & Rehder, 1938 [1]). Considering the multinational shipping history of Hawaiʻi, it is possible many of the established teredinid species in Hawaiʻi are introduced [20]. Therefore, likely more than one long-term brooder in the *L. pedicellatus* complex resides in Hawaiian waters. Given the absence of additional data, we follow the advice provided by Bouchet and Strong [58] and retain these species as synonyms of *L. pedicellatus*. Consequently, *L. reginae* sp. nov. has no synonyms. All characters (morphological and molecular) may be modified with future sampling, aligning with best practices in integrative taxonomy.

## Discussion

Molecular data combined with rigorous exploration of historical collections, extensive sampling, knowledge of life history, and biogeography are necessary to identify and understand the relationships of species within a cryptic species complex [11,59]. Species boundaries inferred from this work reflect congruence among phylogenetic structure, diagnostic molecular characters, and reproductive mode, rather than reliance on any single algorithmic output or genetic distance threshold. This approach has more than doubled the number of species in the *Lyrodus pedicellatus* complex from three to seven species. The high number of cryptic taxa found here, despite limited geographic sampling, is a strong indication that the *L. pedicellatus* complex contains more undiscovered cryptic species, as pallet morphology can be conserved across highly divergent lineages.

*Lyrodus reginae* sp. nov. shares the same pallet morphology and reproductive mode (SqLTB) as *L. pedicellatus*. *Lyrodus reginae* sp. nov. was found on the west side of Hawaiʻi Island in natural wood and recruited to test panels on Oʻahu at Waiʻanae (west side) and in Pearl Harbor (south side). The leeward (west) sides of the islands receive less rain and have few wood resources due to reduced tree and brush density, with few watersheds to carry wood to the ocean. Snorkel surveys yielded abundant submerged wood along east and south coasts of the islands where there are more streams and abundant woody flora. Surveys at west side sites yielded little wood, except for locations near or in estuaries, and/or with overhanging trees. The pediveliger larvae of LTBs can settle immediately after spawning, allowing them to bore into the same wood item as their parent. This may give them an advantage in locations with wood scarcity, as the larvae of STB and oviparous species require 2–6 weeks in the water column before settling [15,60].

We found two established sympatric cryptic species in the Hawaiian Islands, *Lyrodus reginae* sp. nov.*,* a SqLTB, and *L.* cf. *floridanus*, a STB. The STB, found at one location in 1940, was established at multiple locations on three islands by the time our study commenced in 2014. The genetic distance between these two indicates their divergence occurred millions

of years ago. These findings suggest one of two possible scenarios: both are introduced, or the SqLTB is endemic, having evolved in the Hawaiian archipelago, and the STB is introduced. If both were endemic, human shipping would have dispersed the STB long before Edmondson conducted his surveys.

Edmondson [27] found the SqLTB member of the *Lyrodus pedicellatus* complex in abundance, whereas our more extensive survey over a longer period produced only a few specimens with similar morphology and larval brooding strategy. It is unlikely, but possible, that the SqLTB found by Edmondson is a different species than *L. reginae* sp. nov. Edmondson's 1939−1941 survey found a '*L. pedicellatus*' STB (referred to herein as *L.* cf. *floridanus*) at one location only; the port of Hilo on Hawai'i Island. However, our examination of specimens of *L. pedicellatus* morphology in MCZ:Mala:350275, collected in 1978, found the gills packed with D-stage larvae, confirming them as STBs. Thus *L.* cf. *floridanus* was present on O'ahu by 1978. We have since found *L.* cf. *floridanus* widely distributed across Hawai'i Island, Kaua'i, and O'ahu, occurring at a total of 31 sites (34%), including 9 sites in Pearl Harbor. These locations were all protected from open coastal water action (estuaries, ports, and marinas, or enclosed by outer reefs). Regardless of which SqLTB species of the *L. pedicellatus* complex Edmondson collected, our current survey found the abundance and distribution of the SqLTB dramatically decreased, whereas *L.* cf. *floridanus* has spread throughout the islands. This supports the hypothesis that *L.* cf. *floridanus* was introduced at Hilo in the mid-20$^{th}$ century and dispersed throughout the islands, while during the same time period, *L. reginae* sp. nov. declined in abundance and range. This raises the possibility that competition between the two species may have driven these changes.

Our phylogenetic analysis found three SqLTB species of the *Lyrodus pedicellatus* complex. The 'original' *L. pedicellatus,* a SqLTB, established in the northeast Atlantic, was found in San Francisco Bay, California, USA, along with an undescribed SqLTB referred to herein as *L.* cf. *pedicellatus.* The latter is possibly the new species predicted by Borges and Merckelbach [10]. The specimens of these two cryptic species came from the same piece of wood. This is not surprising considering the extensive shipping history of this large, international port, and the subsequent introduction of numerous invertebrate species from around the globe [61]. *Lyrodus* cf. *pedicellatus* groups closely with *L. pedicellatus* itself (Fig 7). The native provenance of this cryptic San Francisco SqLTB species may never be known unless there is a fortuitous collection at its area of origin with subsequent molecular analysis.

Three members of the *Lyrodus pedicellatus* complex are STBs: *L. mersinensis* (Mediterranean), *L.* cf. *mersinensis* (Japan) and *L.* cf. *floridanus* (Hawai'i). The single pallet of the *L. floridanus* (type location: Tampa, Florida, USA) syntype is missing and no reproductive information was provided by Bartsch [62]. Calloway and Turner [16] did not deposit any brooding specimens complete with pallets which could have provided more information on this resurrected species. A thorough survey in Florida for *Lyrodus pedicellatus*-like specimens and molecular data are needed to disentangle this species' complicated status. As such, we refer to our specimens of the Hawaiian STB as *L.* cf. *floridanus*.

*Lyrodus mersinensis* (type locality: Mersin, Turkey) and the STB sequenced from Japan (Amakusa, Kumamoto Pref.) group together as separate species, and sister taxa in a well-supported clade (Fig 7). Librando-Descallar *et al*. [57] suggested the presence of *L. mersinensis* in Japanese waters based on 28S sequences, but did not provide the sequences, molecular analysis, or reproductive information of their specimens. Our analysis used COI, a locus better suited for species-level delimitation, and the bPTP species delimitation analysis supports the Japanese specimen and *L. mersinensis* as two distinct species. We retain the referral of the STB Japanese specimen to *L.* cf. *mersinensis* until further study can resolve its identity. The reproductive status of the seventh species, *Lyrodus* sp. (Florida), is unknown, which leaves its affinities to the resurrected *L. floridanus* of Calloway and Turner [16] uncertain.

This study is the first to address whether reproductive strategy is a response to environmental conditions or a genetically inherited trait within a teredinid cryptic species complex. The three SqLTBs were clearly delimited as separate species from the three STBs and observed brooding mode was consistent, without exception, within each of these phylogenetic species. The analysis herein clearly shows that reproductive strategy (brooding mode) is a genetic trait within the *L. pedicellatus* complex.

Until recently, *Lyrodus pedicellatus* was regarded as one of the most cosmopolitan shipworm species, reported across five continents: in the Atlantic, Pacific, Indian Oceans and the Mediterranean, Caribbean, Black and Red Seas [1,56,63–68]. Our results indicate that dispersal followed by genetic isolation resulted in multiple cryptic species with indistinguishable pallet morphology and two different reproductive strategies. Ocean currents disperse shipworms to new locations via driftwood and larvae. Driftwood carries shipworms from shore to shore, even across ocean basins. Seismic events and tsunamis can greatly amplify this process by increasing the quantity of wood entering the oceans and propelling it across vast distances, enabling transoceanic dispersal [14,29,69]. Most drift logs originating from beyond the Hawaiian Archipelago arrive from the northeast Pacific, with smaller numbers from Asia, the southwest Pacific, and the Pacific coast of South America [21, NCT per. obs.]. Shipworms rafted in driftwood to Hawaiʻi could have established populations prior to human contact, allowing for the evolution of endemic species. Further sampling is needed to fully resolve the relationship and evolution of shipworms across the Pacific.

Vessels from all over the world came to Hawaiʻi, bringing adult shipworms in their hulls and shipworm larvae in ballast water, consequently introducing non-native species [18,20]. If *Lyrodus reginae* sp. nov. arrived in Hawaiʻi via shipping, its geographic origin is unknown, a common pattern among introduced species first described from outside their native regions [20]. We note, for example, both the turbellarian *Taenioplana teredinid* Hyman, 1944 (a shipworm predator), and the copepod *Teredicola typica* Wilson, 1942, (a shipworm parasite) were first described from the Hawaiian Islands, and are now regarded as native to the southern hemisphere [20,70]. The question of *L. reginae* sp. nov.'s origin, native or introduced, can only be resolved through extensive taxonomic and geographic sampling coupled with integrative analysis that combines genetics, anatomy, and life history traits of species within the genus *Lyrodus*. Until that time, we regard it as a cryptogenic species.

It is likely that more cryptic species, both short- and long-term brooders, will be discovered within this complex due to the worldwide distribution of the morphospecies. The discovery of *L. reginae* sp. nov. highlights the effectiveness of an integrative taxonomic approach that encompasses genetic, morphological, and reproductive characters, analysis of bio-geographic patterns and archival material, for taxonomic revisionary work within Teredinidae.

## Conclusion

Herein we demonstrate the presence of seven species within the *Lyrodus pedicellatus* complex, including the new species *Lyrodus reginae* sp. nov., using an integrative approach combining extensive field sampling, morphology, reproductive biology, life history data, molecular data, phylogenetic analysis, and examination of museum material. Genetic analysis revealed reproductive mode is an inherited genetic trait within this complex. We describe *L. reginae* sp. nov., a sequential long-term brooder, differentiated from the other species in this complex by molecular diagnostic characters, and its long-term brooding strategy compared with the short-term brooders *L. mersinensis, L. floridanus, L.* cf. *floridanus, and L.* cf. *mersinensis*.

No new species of teredinid were described from 1979 to 2004. A renewed attention to this family in the last two decades produced seven new species [4,9–12,71,72]. This constitutes a remarkable increase in diversity and supports the need for substantial taxonomic revisionary work to clarify the evolution and biogeography of this bivalve family.

## Acknowledgments

This project was accomplished through the assistance of people and institutions who generously gave their time, expertise, and resources. We are grateful for the assistance of the Malacology and Invertebrate Zoology collections staff and those in the PCMB and the Ethnology Department at the BPBM. We are grateful for the assistance provided by the Amakusa Marine Biology Laboratory, Hawaiʻi Department of Land and Natural Resources Division of Aquatic Resources, U.S. Navy FAC PAC, Hawaiian Institute of Marine Biology, University of Hawaiʻi at Mānoa and Hilo, U.S. Fish and Wildlife Services, Kauaʻi Surf Riders, Hawaiian Malacology Society, National and Oceanic Atmospheric Administration, California

Academy of Sciences, Museum für Naturkunde, Museum of Comparative Zoology, Muséum National d'Histoire Naturelle, National Museum of Nature and Science, and the Smithsonian National Museum of Natural History. We extend our thanks to S. Arakai, A. Baldinger, C. Berg, H. Bolick, J. Boord, B. Buge, P. Bouchet, J. Culliney, T. Davidson, K. Fuller, J. Glazner, J. Goggins, K. Goodale, J. Hafner, T. Haga, S. Hanser, V. Heros, J. Johansen, W. Kawamoto, J.R. Kim, L. Kools, R. Kosaki, D. Lager, J. Lemus, N. Maximenko, S. McCubbins, K. McDermid, A. Nishimoto, J. Pfeiffer, J. Plissner, N. Puillandre, S. Royer, T. Sakihara, R. Springer, T. White, N. Walvoord, N.W. Yeung, and C. Zorn. We thank Dr. James T. Carlton for his guidance throughout the project, and Dr. Alex Strachan and Mr. Glenn Harper from the Plymouth Electron Microscopy Centre. We extend our appreciation to Regina Kawamoto for her constant assistance and support. We thank G. Giribet and an anonymous reviewer for their helpful suggestions and improvements to the manuscript.

## Author contributions

**Conceptualization:** Nancy C. Treneman, Kelli L. DeLeon, J. Reuben Shipway, Luísa M. S. Borges, Kenneth A. Hayes.

**Data curation:** Nancy C. Treneman, Kelli L. DeLeon, Kenneth A. Hayes.

**Formal analysis:** Nancy C. Treneman, Kenneth A. Hayes.

**Funding acquisition:** Nancy C. Treneman.

**Investigation:** Nancy C. Treneman, Kelli L. DeLeon, J. Reuben Shipway, Luísa M. S. Borges, Kenneth A. Hayes.

**Methodology:** Nancy C. Treneman, Kelli L. DeLeon, J. Reuben Shipway, Luísa M. S. Borges, Kenneth A. Hayes.

**Project administration:** Nancy C. Treneman, Kenneth A. Hayes.

**Resources:** Nancy C. Treneman, Kenneth A. Hayes.

**Software:** Kenneth A. Hayes.

**Supervision:** Nancy C. Treneman, Kenneth A. Hayes.

**Validation:** Nancy C. Treneman, Kelli L. DeLeon, J. Reuben Shipway, Luísa M. S. Borges, Kenneth A. Hayes.

**Visualization:** Nancy C. Treneman, Kelli L. DeLeon, J. Reuben Shipway, Luísa M. S. Borges, Kenneth A. Hayes.

**Writing – original draft:** Nancy C. Treneman, Kelli L. DeLeon, J. Reuben Shipway, Luísa M. S. Borges, Kenneth A. Hayes.

**Writing – review & editing:** Nancy C. Treneman, Kelli L. DeLeon, J. Reuben Shipway, Luísa M. S. Borges, Kenneth A. Hayes.

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
