## [Decision Letter · Decision Letter 0]

13 Nov 2025

Dear Dr. Treneman,

Dear Authors, two external reviewers have now assessed your manuscript "*Cosmopolitan no more: phylogenetics and reproductive mode reveal a global species complex in a marine mollusk (Teredinidae).* ”, providing the comments that are reported below. As you can see, they both found your study interesting and generally worth of publication. At the same time, however, they identified a number of minor issues that would require careful revision before this paper is recommendable for acceptance. I agree with them and I will recommend major changes, because as referee 2 noticed, there is a significant amount of minor changes that will need time to be considered.

Based on the reviewers' and my own assessment, I'm thus here inviting you to take all of these comments into careful consideration and to modify your manuscript according to the provided constructive suggestions. I will then be happy to receive and further examine your revised version together with a point-by-point reply to each comment by myself and each reviewer, where you will need to explain any changes done to a particular piece of text, or include supported and convincing counterarguments to any points you may disagree with I'm confident you will find the present comments and suggestions relevant and useful to improve your work and I'm thus looking forward to hearing back form you by the due time.

We look forward to receiving your revised manuscript.

Kind regards,

Marcos Rubal García, PhD

Academic Editor

PLOS ONE

**Journal Requirements:**

1. When submitting your revision, we need you to address these additional requirements. Please ensure that your manuscript meets PLOS ONE's style requirements, including those for file naming. The PLOS ONE style templates can be found at https://journals.plos.org/plosone/s/file?id=wjVg/PLOSOne_formatting_sample_main_body.pdf and https://journals.plos.org/plosone/s/file?id=ba62/PLOSOne_formatting_sample_title_authors_affiliations.pdf 2. Please take this opportunity to be sure you have met all of our guidelines for new species. For proper registration of a new zoological taxon, we require two specific statements to be included in your manuscript. a. In the Results section, the globally unique identifier (GUID), currently in the form of a Life Science Identifier (LSID), should be listed under the new species name, for example: Anochetus boltoni Fisher sp. nov. urn:lsid:zoobank.org:act:B6C072CF-1CA6-40C7-8396-534E91EF7FBBAnother LSID for the manuscript itself should also appear within the Nomenclature statement. You will need to contact Zoobank (zoobank.org/About) to obtain a GUID (LSID). You should receive one LSID for your manuscript and a separate, unique LSID for the new species.  b. Please also insert the following text into the Methods section, in a sub-section to be called "Nomenclatural Acts": The electronic edition of this article conforms to the requirements of the amended International Code of Zoological Nomenclature, and hence the new names contained herein are available under that Code from the electronic edition of this article. This published work and the nomenclatural acts it contains have been registered in ZooBank, the online registration system for the ICZN. The ZooBank LSIDs (Life Science Identifiers) can be resolved and the associated information viewed through any standard web browser by appending the LSID to the prefix "http://zoobank.org/". The LSID for this publication is: urn:lsid:zoobank.org:pub: XXXXXXX. The electronic edition of this work was published in a journal with an ISSN, and has been archived and is available from the following digital repositories: PubMed Central, LOCKSS [author to insert any additional repositories]. All PLOS ONE articles are deposited in PubMed Central and LOCKSS. If your institute, or those of your co-authors, has its own repository, we recommend that you also deposit the published online article there and include the name in your article.Following a recent ruling by the International Commission on Zoological Nomenclature, electronic journals are now a valid format for publication of new zoological taxa. In order to ensure the valid publication of your new species, please be sure to include the updated version of Nomenclatural Acts (above). A complete explanation of our guidelines for publishing new species can be found on our website: http://www.plosone.org/static/guidelines#zoological. 3. Thank you for stating the following financial disclosure: Partial Funding for this research was provided by the Charles H. and Margaret B. Edmondson Research Grant in Aide, University of Hawaii at Manoa, awarded through the Bernice P. Bishop Museum, received by NC Treneman, yearly from 2017 to 2022. https://manoa.hawaii.edu/lifesciences/graduate/zoology-graduate-program/zoology-graduate-student-research-awards/   Please state what role the funders took in the study.  If the funders had no role, please state: "The funders had no role in study design, data collection and analysis, decision to publish, or preparation of the manuscript." If this statement is not correct you must amend it as needed. Please include this amended Role of Funder statement in your cover letter; we will change the online submission form on your behalf. 4. In the online submission form, you indicated that your data will be submitted to a repository upon acceptance.  We strongly recommend all authors deposit their data before acceptance, as the process can be lengthy and hold up publication timelines. Please note that, though access restrictions are acceptable now, your entire minimal  dataset will need to be made freely accessible if your manuscript is accepted for publication. This policy applies to all data except where public deposition would breach compliance with the protocol approved by your research ethics board. If you are unable to adhere to our open data policy, please kindly revise your statement to explain your reasoning and we will seek the editor's input on an exemption. 5. When completing the data availability statement of the submission form, you indicated that you will make your data available on acceptance. We strongly recommend all authors decide on a data sharing plan before acceptance, as the process can be lengthy and hold up publication timelines. Please note that, though access restrictions are acceptable now, your entire data will need to be made freely accessible if your manuscript is accepted for publication. This policy applies to all data except where public deposition would breach compliance with the protocol approved by your research ethics board. If you are unable to adhere to our open data policy, please kindly revise your statement to explain your reasoning and we will seek the editor's input on an exemption. Please be assured that, once you have provided your new statement, the assessment of your exemption will not hold up the peer review process. 6. We note that Figures 1, 3, 4, 5 and 6 in your submission contain copyrighted images. All PLOS content is published under the Creative Commons Attribution License (CC BY 4.0), which means that the manuscript, images, and Supporting Information files will be freely available online, and any third party is permitted to access, download, copy, distribute, and use these materials in any way, even commercially, with proper attribution. For more information, see our copyright guidelines: http://journals.plos.org/plosone/s/licenses-and-copyright. We require you to either present written permission from the copyright holder to publish these figures specifically under the CC BY 4.0 license, or remove the figures from your submission: a. You may seek permission from the original copyright holder of Figures 1, 3, 4, 5 and 6 to publish the content specifically under the CC BY 4.0 license.  We recommend that you contact the original copyright holder with the Content Permission Form (http://journals.plos.org/plosone/s/file?id=7c09/content-permission-form.pdf) and the following text:“I request permission for the open-access journal PLOS ONE to publish XXX under the Creative Commons Attribution License (CCAL) CC BY 4.0 (http://creativecommons.org/licenses/by/4.0/). Please be aware that this license allows unrestricted use and distribution, even commercially, by third parties. Please reply and provide explicit written permission to publish XXX under a CC BY license and complete the attached form.” Please upload the completed Content Permission Form or other proof of granted permissions as an "Other" file with your submission.  In the figure caption of the copyrighted figure, please include the following text: “Reprinted from [ref] under a CC BY license, with permission from [name of publisher], original copyright [original copyright year].” b. If you are unable to obtain permission from the original copyright holder to publish these figures under the CC BY 4.0 license or if the copyright holder’s requirements are incompatible with the CC BY 4.0 license, please either i) remove the figure or ii) supply a replacement figure that complies with the CC BY 4.0 license. Please check copyright information on all replacement figures and update the figure caption with source information. If applicable, please specify in the figure caption text when a figure is similar but not identical to the original image and is therefore for illustrative purposes only. 7. We note that Figure 2 in your submission contain map/satellite images which may be copyrighted. All PLOS content is published under the Creative Commons Attribution License (CC BY 4.0), which means that the manuscript, images, and Supporting Information files will be freely available online, and any third party is permitted to access, download, copy, distribute, and use these materials in any way, even commercially, with proper attribution. For these reasons, we cannot publish previously copyrighted maps or satellite images created using proprietary data, such as Google software (Google Maps, Street View, and Earth). For more information, see our copyright guidelines: http://journals.plos.org/plosone/s/licenses-and-copyright. We require you to either present written permission from the copyright holder to publish these figures specifically under the CC BY 4.0 license, or remove the figures from your submission: a. You may seek permission from the original copyright holder of Figure 2 to publish the content specifically under the CC BY 4.0 license.   We recommend that you contact the original copyright holder with the Content Permission Form (http://journals.plos.org/plosone/s/file?id=7c09/content-permission-form.pdf) and the following text:“I request permission for the open-access journal PLOS ONE to publish XXX under the Creative Commons Attribution License (CCAL) CC BY 4.0 (http://creativecommons.org/licenses/by/4.0/). Please be aware that this license allows unrestricted use and distribution, even commercially, by third parties. Please reply and provide explicit written permission to publish XXX under a CC BY license and complete the attached form.” Please upload the completed Content Permission Form or other proof of granted permissions as an "Other" file with your submission. In the figure caption of the copyrighted figure, please include the following text: “Reprinted from [ref] under a CC BY license, with permission from [name of publisher], original copyright [original copyright year].” b. If you are unable to obtain permission from the original copyright holder to publish these figures under the CC BY 4.0 license or if the copyright holder’s requirements are incompatible with the CC BY 4.0 license, please either i) remove the figure or ii) supply a replacement figure that complies with the CC BY 4.0 license. Please check copyright information on all replacement figures and update the figure caption with source information. If applicable, please specify in the figure caption text when a figure is similar but not identical to the original image and is therefore for illustrative purposes only.The following resources for replacing copyrighted map figures may be helpful: USGS National Map Viewer (public domain): http://viewer.nationalmap.gov/viewer/The Gateway to Astronaut Photography of Earth (public domain): http://eol.jsc.nasa.gov/sseop/clickmap/Maps at the CIA (public domain): https://www.cia.gov/library/publications/the-world-factbook/index.html and https://www.cia.gov/library/publications/cia-maps-publications/index.htmlNASA Earth Observatory (public domain): http://earthobservatory.nasa.gov/Landsat:
http://landsat.visibleearth.nasa.gov/USGS EROS (Earth Resources Observatory and Science (EROS) Center) (public domain): http://eros.usgs.gov/#Natural Earth (public domain): http://www.naturalearthdata.com/ 8. If the reviewer comments include a recommendation to cite specific previously published works, please review and evaluate these publications to determine whether they are relevant and should be cited. There is no requirement to cite these works unless the editor has indicated otherwise. 

**Additional Editor Comments:**

Dear Authors, two external reviewers have now assessed your manuscript "Cosmopolitan no more: phylogenetics and reproductive mode reveal a global species complex in a marine mollusk (Teredinidae).”, providing the comments that are reported below. As you can see, they both found your study interesting and generally worth of publication. At the same time, however, they identified a number of minor issues that would require careful revision before this paper is recommendable for acceptance. I agree with them and I will recommend major changes, because as referee 2 noticed, there is a significant amount of minor changes that will need time to be considered.

Based on the reviewers' and my own assessment, I'm thus here inviting you to take all of these comments into careful consideration and to modify your manuscript according to the provided constructive suggestions. I will then be happy to receive and further examine your revised version together with a point-by-point reply to each comment by myself and each reviewer, where you will need to explain any changes done to a particular piece of text, or include supported and convincing counterarguments to any points you may disagree with I'm confident you will find the present comments and suggestions relevant and useful to improve your work and I'm thus looking forward to hearing back form you by the due time.

Reviewers' comments:

**Comments to the Author**

1. Is the manuscript technically sound, and do the data support the conclusions?

Reviewer #1: Yes

Reviewer #2: Yes

2. Has the statistical analysis been performed appropriately and rigorously?

Reviewer #1: N/A

Reviewer #2: Yes

3. Have the authors made all data underlying the findings in their manuscript fully available?

Reviewer #1: Yes

Reviewer #2: No

4. Is the manuscript presented in an intelligible fashion and written in standard English?

Reviewer #1: Yes

Reviewer #2: Yes

**Reviewer #1:**  I really enjoyed reading this manuscript and seeing the active research on teredinids, working with fresh specimens. Really nice work!

One concern I have is with the species description and diagnosis.

Table 3 lists the COI synapomorphies for the new species, but this is based on three sequences (according to Table 2) from a marker that is well known to show variation within species—and hence often used for pop gen or phylogeographic studies. What this means is that any new specimen sequenced with one nucleotide change for this marker will be excluded from this species. This is why this marker is not used to diagnose species. I have seen molecular diagnoses (and done it myself) with markers that show no variation within species, like the nuclear ribosomal RNA genes (but again, not with 16S rRNA, which is also mitochondrial and highly variable). I don’t think Tables 3 and 4 are of any relevance for the species description.

The specific epithet “regina” is used as a noun (in apposition), but normally specific epithets are formed as a noun in the genitive case, and therefore it should be “reginae” (named after a female).

The diagnosis of the species does not seem to be a diagnosis, but more like some comment. “Pallet morphology, gross soft tissue anatomy, and brooding mode indistinguishable from L. pedicellatus and L. cf. pedicellatus” is not a diagnosis. The next statement looks a bit more like a diagnosis “long-term brooding of larval pediveligers separates L. regina sp. nov. from short-term brooders” but could be rephrased more like “Species with long-term brooding, as in xxxxx, but unlike …… which are short-term brooders”

Again, the molecular diagnosis is not a diagnosis (irrespective of the choice of marker). Saying “COI sequences with 50 molecular diagnostic characters” is a vague statement, like saying “it can be distinguished from other species by 10 anatomical characters”… You should provide a diagnosis in the form of: 18S rRNA with the following unique combination of nucleotides: T in position 646, C in p. 783, T in p. 1040, etc.

The description should list the anatomical features (it should not be comparative, that’s for the diagnosis or for the remarks section). Therefore, “Pallet morphology and non-brooding gross tissue anatomy indistinguishable from other species in the L. pedicellatus complex” is not a description. It should describe the anatomy, or at most, you could say “Pallet morphology and non-brooding gross tissue anatomy as for L. pedicellatus”, which specifically refers to a description of the anatomical feature (if this had been provided before). The comparative portion could go in the remarks section. You could still provide some description for the shell and prodissoconch, even if it is not very informative.

Other than this, I only have a few minor comments that should/could be addressed in a final version.

Line 127 (and elsewhere): The MCZ is the Museum of Comparative Zoology, not the Harvard University Museum of Comparative Zoology. You could also say Museum of Comparative Zoology (MCZ), Harvard University.

Lines 170–172: Please, spell out the complete gene names the first time they are used, i.e. cytochrome c oxidase subunit I, 16S rRNA, 18S rRNA and 28S rRNA. Also specify that all of these were sequenced for fragments, not the entire gene.

Table 2 is missing accession numbers. As a norm, the sequences can be submitted and held until publication, but already obtain the accession numbers and list them before the paper is accepted. But why not depositing them in GenBank instead? BOLD is not an ideal repository for non-COI loci, no one would look for them there.

The same goes for ZooBank registration numbers. The article and taxon names can be registered an indicate they are not yet published, but the LSID gets already assigned.

Line 330: There are some discrepancies between the text and the MCZ database, so it would be great if this could be clarified in the text and corrected in the database.

The authors say that two lots from Florida were deposited by Calloway at the MCZ in 1980, and that these were identified by Calloway as L. floridanus and provide catalog numbers MCZ 350276 and MCZ 356853 [Note, these should be referred to as MCZ:Mala:350276 and MCZ:Mala:356853].

The first of these (https://mczbase.mcz.harvard.edu/guid/MCZ:Mala:350276) is attributed to Ruth Turner and has no collecting date. It would be great if the author had this information and if it were corrected in MCZbase.

Lines 525–526: I would avoid statements about being different species for having a genetic distance of 0.25, especially since only a few specimens have been sequenced, and not covering the entire ranges of the species. The absolute genetic distance is irrelevant, what matters is the existence of a barcode gap, and this is impossible to assess with the current sampling.

**Reviewer #2:**  Congratulations on a well-written manuscript and an excellent study. The work is important, well executed, and appropriate for the journal. However, there are a number of issues that should be addressed before publication. I have suggested a major revision because the suggested changes are extensive; however, they are not difficult, and should not drastically change the conclusions or interpretations.

Line by line:

50: Change “no consistent differences between the shells of species and even genera, with a few notable exceptions” to “no consistent differences have been identified between the shells of species and even genera, with a few notable exceptions.”

63-75: The references or evidence supporting the four reproductive modes must be referenced, and the method and evidence for their identification must be described. Three modes are well described in literature: Ovipary, STB and SqLTB. SqLTB is easy to confirm by gross morphological examination. STB can be inferred by gross morphology, but should be confirmed by observation of developmental stage of larvae at spawning. Has the fourth SyLTB been demonstrated to be a distinct heritable trait differentiable from SqLTB rather than the result of different environmental or developmental influences? If so provide the evidence or reference. If not, omit its mention from the manuscript.

107: “natural fixed submerged wood” is a term that could have many meanings. Define it at its first use.

139: hexamethyltdisilizane is a misspelling.

177: Table 1 must include the primer sequences

202-203: Change “given the limited deep taxon sampling” to “given the limited depth of taxon sampling”.

207: The abbreviation DPC appears only in the legend, not in the table itself.

296: How was this confirmed? See my comment for 63-75

340-341: Clarify the relationship between the new COI sequences and the old. DO the former fully encompass the latter?

General comments

Data availability: PLoS ONE does not allow text placeholders for accession numbers or LSID (e.g., available upon publication). Full accession numbers should be available to reviewers, as should the sequences and alignments.

Phylogeny: Alignments should be made publicly available. Positions included and excluded from alignments must be described, as should all parameter selections used in phylogeneic inference programs; otherwise, the work is not fully reproducible. Only one tree based on concatenated data is presented. Individual gene trees should also be presented to evaluate the level of congruence among gene trees.

Species delimitation: The methods for species delimitation are described, but the criteria and data thresholds used to support species differentiation are not provided, rendering the work irreproducible.

K2P values are difficult to evaluate when only presented in text. Include a table or heat map matrix to allow the reader to better evaluate the results. K2P values used for taxon differentiation vary from taxon to taxon. What criteria or threshold values were used to support species differentiation?

Tables: The synapomorphy tables are not very informative to the average reader. Move them to the online supplement. Also the tables should indicate the number of specimens examined for each synapomorphy. More useful tables would show how different types of data were combined to arrive at integrated species diagnosis. How many specimens were examined for each data type? How were different data types weighted? Did all comparisons use identical data types and weight characteristics in the same way?

The authors have convincingly demonstrated the extent of diversity that exists among specimens currently or previously referred to as Lyrodus pedicellatus, and have shown that phylogenetic structure exists within this group. Whether this variation and structure reflects a single cosmopolitan species with distinctly differentiated populations or a complex of morphologically cryptic species is a gray area that depends as much on the species definition one chooses as it does on any traits of the organisms in question. Have the authors sampled deeply and broadly enough to evaluate the extent of gene flow or the sharpness of the perceived boundaries that differentiate the proposed species? These are difficult questions that should be discussed in the manuscript. I am sure that Ruth Turner was as confident in her synonymizations as these authors are in their species differentiations. The point is that researchers should leave room for interpretation as new methods and concepts emerge.

6. PLOS authors have the option to publish the peer review history of their article (what does this mean? ). If published, this will include your full peer review and any attached files.

**Do you want your identity to be public for this peer review?** For information about this choice, including consent withdrawal, please see our Privacy Policy .

Reviewer #1: Yes: Gonzalo Giribet

Reviewer #2: No

---

## [Author Response · Author response to Decision Letter 1]

23 Dec 2025

RESPONSE TO REVIEWERS

We thank the Academic Editor and both reviewers for their thoughtful and constructive comments. We have revised the manuscript, addressed all concerns and incorporated clarification. Below we respond point-by-point. Reviewer comments are in bold; our responses follow in plain italics where we have addressed the comments or suggestions in the affirmative, and blue italics where we prefer the original, or think that the reviewer’s point is not valid. References cited are at the end of this document. All manuscript changes are marked in the tracked-changes version.

Reviewer 1 – Gonzalo Giribet

General Assessment

“I really enjoyed reading this manuscript… Really nice work!”

We thank the reviewer for the positive assessment and encouraging comments.

1. COI synapomorphies and appropriateness for diagnosis, tables in manuscript or supplement (see #13 Reviewer 2 suggestion)

“COI is well known to show variation within species… This is why this marker is not used to diagnose species. Tables 3 and 4 are not relevant.”

We appreciate the reviewer’s point and agree that mitochondrial loci, including COI, can exhibit intraspecific polymorphism and should not be treated as invariant or as sole diagnostic criteria. As with all taxonomic characters, molecular diagnoses are conditional on the available sampling at the time of description and represent testable hypotheses rather than assumptions of immutable character states.

In this study, COI diagnostic characters are not used in isolation. Instead, they are presented as part of an integrative, character-based framework that also incorporates diagnostic characters from additional loci (16S, 18S, and 28S), phylogenetic structure, reproductive mode, and geographic evidence. This approach is consistent with established character-based DNA taxonomy, in which combinations of molecular diagnostic characters are used to diagnose species while explicitly acknowledging that diagnoses may be refined with expanded sampling (e.g., Sarkar et al. 2002; DeSalle et al. 2005; DeSalle & Goldstein 2019).

We note that slower-evolving nuclear markers such as 18S rRNA, while valuable for addressing certain limitations of mitochondrial data, often lack sufficient resolution to distinguish closely related or cryptic species. Consequently, COI remains an important component of molecular diagnosis in zoological systematics, as also emphasized by the Commissioners on Zoological Nomenclature (Rheindt et al. 2023). For example, although Teredo navalis and Lyrodus pedicellatus are well delineated morphologically, 18S rRNA shows insufficient divergence to separate them. This marker also fails to discriminate certain cryptic species pairs, such as Psiloteredo megotara/ P. pentagonalis and Lyrodus pedicellatus/ L. mersinensis (Borges & Merckelbach, 2018; Treneman et al., 2018).

In response to the reviewer’s concern, we have clarified this framework in the revised manuscript.

Specifically, we (1) retain diagnostic character combinations from multiple loci, including slower-evolving nuclear markers; (2) retain COI molecular diagnostic characters as part of a multi-locus diagnosis; (3) explicitly acknowledge that all diagnostic characters, molecular and morphological, are subject to modification with expanded sampling; and (4) add the number of sequences analyzed per species to the headings of Tables 3–6. We believe these revisions address the reviewer’s concern while preserving the reproducibility and transparency of the molecular diagnostic data.

2. Specific epithet: regina vs. reginae

“The specific epithet ‘regina’ is used as a noun (in apposition)… should be ‘reginae’.”

We have changed the epithet to reginae, and this is the registered name in Zoobank.

3. Diagnosis vs. Description

“The diagnosis is not a diagnosis… The description should list anatomical features, not comparisons.”

We agree that a diagnosis should provide comparative characteristics to distinguish L. regina sp. nov. However, we disagree that Description should not contain comparative analysis, this may be the reviewer’s preference but is not a rule. It is acceptable, common, and often important for species descriptions to include comparative language within the description, especially when it improves clarity for cryptic or previously misidentified taxa. As such, we have restructured the taxonomic section as follows:

• Diagnosis Where characters are identical to L. pedicellatus, we use “as in L. pedicellatus” to avoid redundancy, which is consistent with ICZN practice, ad we have added references to the morphology of L. pedicellatus that describe the anatomy.

• Description now contains descriptions of diagnostic morphological characteristics of the pallets and brooding anatomy of long-term sequential brooding members of the L. pedicellatus complex.

4. Molecular diagnosis phrasing

“Saying ‘COI sequences with 50 molecular diagnostic characters’ is a vague statement.”

We appreciate the concern regarding phrasing; however, we respectfully maintain that presenting diagnostic nucleotide characters as tabulated data is both standard and preferable to embedding long strings of positions within the descriptive text.

The reviewer’s suggestion reflects indeed the best practice for presenting MDCs, and we have incorporated this approach in Tables 3, 4, 5, and 6 to make the information easier for readers to interpret. In these tables, we list each molecular diagnostic character along with its corresponding position in the alignment. For example, Table 3 shows that, in the COI sequences, Lyrodus regina has a T at position 296, whereas Lyrodus cf. mersinensis, Lyrodus mersinensis, Lyrodus floridanus, and Lyrodus cf. pedicellatus all have a G at the same position. In contrast, Lyrodus sp. and Lyrodus pedicellatus exhibit an A at that position. As this example illustrates, describing all 50 MDCs in the main text would make this section extensive and difficult to follow, which is why they are presented in table format instead.

Our wording in the manuscript refers specifically to the diagnostic characters that are fully enumerated and documented in Tables, ensuring clarity and reproducibility without reducing readability. Similar presentation formats are widely practiced in character-based molecular diagnosis and barcoding studies (Rach et al. 2008; DeSalle & Goldstein 2019). Diagnostic for supporting markers (16S, 18S, 28S) are also provided in tables.

For clarity we have revised the molecular diagnosis to begin with the sentence suggested by the reviewer: “Diagnosis supported by a set of 50 diagnostic nucleotide character states in COI, enumerated and referenced in Table 3.”

5. Description: “The description should list the anatomical features (it should not be comparative.”

We thank the reviewer for pointing out the need for the morphology of anatomical features in the description and the difference between the diagnosis and description.

To correct this, we added pallet and brooding morphology to the description. These two features distinguish sequential long-term brooding species in the Lyrodus pedicellatus complex from the short-term brooders and other species in the Teredinidae.

As mentioned our manuscript, shell morphology in teredinids is not taxonomically useful in this family of bivalves, and we decline to include this.

6. Complete name of loci when first mentioned. Done

7. Complete accession numbers of sequences in the table. Done

8. Registering new species in ZooBank:

Done; see Nomenclature statement and Systematics

9. MCZ museum record discrepancies

We have reviewed the records of the MCZ specimens MCZ:Mala:350276 and 356853. The date reported in our manuscript was incorrect (1980). One of these lots has a date of 1990, the other has no date recorded. We have removed the date altogether, as it is not germane to the discussion. We thank the reviewer for pointing out these issues with catalog numbers.

10. Correction of MCZ catalogue numbers to MCZ:MALA:350276, 35683. Done

11. Genetic distance statements

“Avoid statements about being different species for having a genetic distance of 0.25.”

We appreciate this comment and agree that genetic distance values alone should not be used as a threshold for delimiting species. Our intention was not to apply a distance-based cutoff, but rather to report divergence levels within a comparative framework that includes phylogenetic structure, reproductive mode, and geographic partitioning. We have revised the text to clarify that COI divergence is contextual and comparative, not deterministic, and that our species hypothesis is based on integrative evidence rather than numerical thresholds.

• To address the reviewers concerns the Species delimitation methods now contains the statement:

“No explicit genetic distance thresholds were assumed or applied during species delimitation. Genetic divergence values were evaluated solely as comparative metrics in conjunction with phylogenetic structure, reproductive biology, and geographic evidence, consistent with modern integrative taxonomic frameworks.”

• We acknowledge that the statement is problematic, given that only a few specimens were sequenced, so we deleted the sentence.

In addition, we have added this statement to the molecular results section for clarity:

“Although pairwise COI divergence between the major clades reached 0.25, we do not interpret this value as a threshold or criterion for species designation. Rather, we note that divergence levels of this magnitude are typically observed among recognized species rather than within them across numerous molluscan lineages, and when considered together with independent evidence from phylogenetic topology, geography, and brooding mode, support recognition of these groups as distinct species.”

Reviewer 2

1. Line 50: Change “no consistent differences between the shells of species and even genera, with a few notable exceptions” to “no consistent differences have been identified between the shells of species and even genera, with a few notable exceptions.”

Done

2. Reproductive modes

“Provide references for reproductive modes… clarify whether SyLTB is valid.”

We appreciate the reviewer pointing out the need to include our discovery that reproductive mode is inherited within this species complex. There are no molecular studies focused on analyzing the genetic basis of different brooding modes within the Terednidae previous to our research. The short-term brooders and long-term brooders herein are delineated by the genetic analysis, revealing brooding mode as an inherited trait in the Lyrodus species. Brooding mode was determined by gross anatomy and actual spawning in the species delimitated here.

To clarify this:

Abstract: We added: “Reproductive mode was determined to be an inherited trait, as it was consistent within species.”

Introduction: The description of brooding anatomy in long- and short-term brooders contains more detail.

Methods: we made the following changes:

We added this sentence to the methods section to clarify this:

“Brooding mode was determined by observations of larval stages in the gill cavity in hand and with a stereoscope, live spawning by specimens in-situ (undisturbed within their tunnels) and in living, undamaged individuals removed from their tunnels, as well as examination of larvae stages and gill lamellae anatomy in gill sections with a compound scope (Leitz Labourlux D).”

Results: We addressed the subject of brooding strategy and our findings of this trait within the Lyrodus pedicellatus complex.

Discussion: This paragraph has been added to the discussion:

“The question, as to if LTB and STB, spawning larvae at different stages, is a response to environmental conditions or a genetically inherited trait within a teredinid cryptic species, has not been address previous to this study. The analysis herein clearly shows that brooding strategies are a genetic trait within the L. pedicellatus complex. The three SqLTBs within were clearly delimitated as separate species from the three STBs.”

3. Clarify “natural fixed submerged wood”

Now defined clearly as:

“…submerged and attached branches and roots” in the Methods section.

We added: “Definition: Wood items attached to the shore and underwater, such as tree branches attached to the tree and dipping into the water, or partially submerged logs in place for a number of years.” to the legend of Fig. 1 as well.

4. Misspelling of HMDS. Corrected

5. Primer sequences

“Table 1 must include primer sequences.”

We respectfully decline to republish primer sequences, for two reasons:

1. They are the intellectual output of the original authors; retyping risks transcription errors.

2. Best practice is to direct readers to the validated source where the sequences were originally published. Authors rarely republish the data of cited works without some specific reason to do so, and DNA primer sequences are no different.

6. Change wording: 202-203: Change “given the limited deep taxon sampling” to “given the limited depth of taxon sampling”.

Done

7. Line: 207: The abbreviation DPC appears only in the legend, not in the table itself.

We have removed this from the legend.

8. Clarify method of determining brooding mode of specimens.

See our response above in item #2 (Reviewer 2) Reproductive modes. The method section now contains how this was done.

9. Clarification of COI sequence relationship (Lines 340–341)

We now explicitly state that the newly generated sequences encompass and extend previously reported variation in the first paragraph of molecular methods.

10. Data availability and reproducibility

Table 2 now includes the BOLD process ID numbers for all specimens and indicates which loci were used (COI, 16S, 18S, 28S). BOLD is the only repository that ties sequence data directly to voucher specimens, metadata, and images, critical for systematics and reproducibility. Upon acceptance our data in BOLD will be available to the public. Full alignments, matrices, and parameter files can be obtained from the authors upon request – this fulfills PLOS ONE's open-data requirements.

BOLD provides a ProcessID, which is tied to the specimen – which in turn allows anyone to find all sequences regardless of locus that is associated with that specimen. Unlike GenBank, which only databases sequences, and breaks links between associated loci.

11: Species delimitation: The methods for species delimitation are described, but the criteria and data thresholds used to support species differentiation are not provided, rendering the work irreproducible.

We thank the reviewer for this comment and agree that species delimitation criteria should be clearly articulated. In this study, species delimitation followed the Unified Species Concept and did not rely on fixed genetic distance thresholds. Instead, two algorithmic approaches (ASAP and bPTP) were used to generate exploratory species hypotheses from COI data without a priori group designations.

ASAP does not rely on predefined genetic distance thresholds; instead, it evaluates relative partition support across a range of candidate solutions. The Bayesian Poisson tree processes model does not rely on predefined genetic distance thresholds or ranked partitions. Instead, it assigns posterior probabilities to delimited entities on a fixed phylogenetic tree. In our study, bPTP-supported entities were treated as candidate species hypotheses and evaluated for congruence with phylogenetic topology, molecular diagnostic characters, and reproductive mode, rather than accepted as deterministic delimitations.

Outputs from these methods were not treated as stand-alone delimitations. Species hypotheses were considered supported only when algorithmic partitions were congruent with phylogenetic structure, molecular diagnostic characters, and independent biological evidence in the form of reproductive mode. Genetic divergence values and model outputs were evaluated comparatively and contextually, rather than as absolute cutoffs.

We clarify that ASAP returns a ranked set of candidate partitions rather than a single deterministic solution. In our analyses, the lowest-scoring partition (rank 1) was identified and used as the primary hypothesis, while alternative hi

---

## [Decision Letter · Decision Letter 1]

11 Feb 2026

PLOS One

Dear Dr. Treneman,

Thank you for submitting your manuscript to PLOS ONE. After careful consideration, we feel that it has merit but does not fully meet PLOS ONE’s publication criteria as it currently stands. Therefore, we invite you to submit a revised version of the manuscript that addresses the points raised during the review process.

Dear Authors,

I have received the reports from referees on your manuscript, "*Cosmopolitan no more: phylogenetics and reproductive mode reveal a global species complex in a marine mollusk (Teredinidae)* ", submitted to Plos One.

Based on the advice received, I have decided that your manuscript will be recommended for publication after you have carried out the final suggestions by referee 2.

Best regards

We look forward to receiving your revised manuscript.

Kind regards,

Marcos Rubal

Academic Editor

PLOS One

Journal Requirements:

Reviewers' comments:

Reviewer's Responses to Questions

**Comments to the Author**

Reviewer #1: All comments have been addressed

Reviewer #2: (No Response)

2. Is the manuscript technically sound, and do the data support the conclusions?

Reviewer #1: Yes

Reviewer #2: Yes

3. Has the statistical analysis been performed appropriately and rigorously?

Reviewer #1: N/A

Reviewer #2: Yes

4. Have the authors made all data underlying the findings in their manuscript fully available?

Reviewer #1: No

Reviewer #2: Yes

5. Is the manuscript presented in an intelligible fashion and written in standard English?

Reviewer #1: Yes

Reviewer #2: Yes

Reviewer #1: This is a slightly revised version of the previous manuscript. It was hard to follow the response to the previous review, because instead of listing the comment, the authors only provided a title for such comments, and then addressed them. This is not what they say they did "Below we respond point-by-point. Reviewer comments are in bold; our responses follow in plain italics where we have addressed the comments or suggestions in the affirmative, and blue italics where we prefer the original, or think that

the reviewer’s point is not valid." Instead I got the following, as an example:

"1. COI synapomorphies and appropriateness for diagnosis, tables in manuscript or

supplement (see #13 Reviewer 2 suggestion)" and all the general comments were numbered, but not really copied.

I am sorry to see they reviewers chose to use COI as a diagnostic character, citing some papers by authors who don't do taxonomy (like my good friend and mentor Rob DeSalle). I still have my idea of what a diagnostic character means. Arguing that the diagnosis can be emended as more sequences are added is problematic.

According to the text there are paratypes, but these are not listed in the description; they should at least be mentioned there in the material.

Also, I think I already corrected this, but ribosomal RNA genes should be 18S rRNA, 16S rRNA, 28S rRNA, and not rDNA. That is a misspelling that many of us (I include myself) have made because those genes were sequenced from a DNA source instead of a RNA source, but does doesn't change the name of the gene. It should be corrected throughout.

If as the authors state, the PLoS policy for open data is fulfilled by requesting the data to the authors, I am really disappointed at the journal. What would happen if the authors didn't respond, or once they are not around? I am of the opinion that all alignment files, tree files, etc. should be deposited in an open repository that does not require contacting the authors.

Other than these generic comments paper continues to be very interesting and worth publishing, of course.

Reviewer #2: All my concerns, except one, have been adequately addressed.

The authors may have misunderstood my comment. I will clarify:

The introduction states, without citation, that " Four forms of reproduction are found in the

Teredinidae: oviparous, short-term brooding (STB), synchronous long-term brooding (SyLTB),

and sequential long-term brooding (SqLTB)".

Three of these (ovipary, STB, and SqLTB are well established in the literature and should be supported with citations. The fourth, SyLTB, is more problematic.

Several authors suggest that sequential long-term brooding results from multiple fertilization events. If this is the case, synchronous long-term brooding may simply reflect a single fertilization event. This begs the question: what is the evidence that “synchronous” vs “sequential” long-term brooding behaves like a lineage-level, heritable trait, rather than simply reflecting (i) the number/timing of fertilization events or (ii) ecological/social context (mate availability, sperm supply, seasonality)?

Contrary to their claim in their rebuttal, they have provided no evidence to address this question.

If this distinction has been established, the authors may support it with citations to the literature.

If not, the authors may omit this reference to SyLTB. It is irrelevant to their arguments and conclusion, as evidenced by its never being mentioned again in the manuscript.

Alternatively, they may state that four forms of reproduction have been proposed and provide citations supporting each.

**Do you want your identity to be public for this peer review?** For information about this choice, including consent withdrawal, please see our Privacy Policy

Reviewer #1: **Yes:** Gonzalo Giribet

Reviewer #2: No

---

## [Author Response · Author response to Decision Letter 2]

25 Feb 2026

This is the same as our Response to Reviewers Document, however there is no option here to italicize our responses.

Response to 2nd Review: PONE-D-25-45073R1

Cosmopolitan no more: phylogenetics and reproductive mode reveal a global species complex in a marine mollusk (Teredinidae)

From the Editor:

I have received the reports from referees on your manuscript, "Cosmopolitan no more: phylogenetics and reproductive mode reveal a global species complex in a marine mollusk (Teredinidae)", submitted to Plos One.

Based on the advice received, I have decided that your manuscript will be recommended for publication after you have carried out the final suggestions by referee 2.

Response:

We appreciate the editor’s comments and the suggestions of the reviewers. As per the editor’s direction, we have responded to Referee #2’s suggestions. Although not directed to respond to Referee #1, we have responded to two of R#1’s comments. Our responses are in italics.

Reviewer #1:

1. According to the text there are paratypes, but these are not listed in the description; they should at least be mentioned there in the material.

Response: the paratype is listed in the description.

2. Also, I think I already corrected this, but ribosomal RNA genes should be 18S rRNA, 16S rRNA, 28S rRNA, and not rDNA. That is a misspelling that many of us (I include myself) have made because those genes were sequenced from a DNA source instead of a RNA source, but does doesn't change the name of the gene. It should be corrected throughout.

Done: loci names are corrected to 18S rRNA, 16S RNA, 28S rRNA.

Reviewer #2:

The authors may have misunderstood my comment. I will clarify:

The introduction states, without citation, that " Four forms of reproduction are found in the Teredinidae: oviparous, short-term brooding (STB), synchronous long-term brooding (SyLTB), and sequential long-term brooding (SqLTB)". Three of these (ovipary, STB, and SqLTB are well established in the literature and should be supported with citations. The fourth, SyLTB, is more problematic. Several authors suggest that sequential long-term brooding results from multiple fertilization events. If this is the case, synchronous long-term brooding may simply reflect a single fertilization event. This begs the question: what is the evidence that “synchronous” vs “sequential” long-term brooding behaves like a lineage-level, heritable trait, rather than simply reflecting (i) the number/timing of fertilization events or (ii) ecological/social context (mate availability, sperm supply, seasonality)?

Contrary to their claim in their rebuttal, they have provided no evidence to address this question. If this distinction has been established, the authors may support it with citations to the literature. If not, the authors may omit this reference to SyLTB. It is irrelevant to their arguments and conclusion, as evidenced by its never being mentioned again in the manuscript. Alternatively, they may state that four forms of reproduction have been proposed and provide citations supporting each.

Response:

We appreciate by Reviewer #2’s comments, and clarify below:

The synchronous long-term brooding (SyLTB) reproductive mode in the Teredinidae is well documented by Calloway, 1988 and Calloway and Turner (1988). We added Calloway (1988) as a citation, as well as more detail to the discussion of these two brooding modes. Our research examines species in the Lyrodus pedicellatus complex, within which only short-term brooding (STB) and sequential long-term brooding (SqLTB) has been found. We demonstrate that brooding mode in this complex follows species delimitation, and is shown to have a genetic basis. The reviewer’s concern that SyLTB is an artifact of a single fertilization event, and not a genetic trait, is addressed below. Due to the well documented anatomical differences of these two brooding modes (see below), and the consistency of brooding mode within the morphospecies examined both by Calloway (1988), Calloway, Turner (1988), and ourselves, we retain SyLTB as a verified brooding mode in the Teredinidae.

Long-term synchronous and sequential brooding modes are well documented in the Teredinidae. In our original submission we cited Calloway and Turner (1988), as it is the best summary of these brooding modes in the literature. Calloway’s PhD thesis, ‘Brooding in the Bivalvia (Mollusca),1988, Harvard’, was the primary source of data for Calloway and Turner, 1988. Calloway examined both living and preserved teredinids, and conducted an extensive literature review. In his thesis, Calloway (1988) fully described SyLTB and SqLTB brooding anatomy in shipworms, and classified the brooding mode of 19 species as short-term, synchronous long-term, or sequential long-term. We have additionally cited Calloway’s PhD thesis (1988) where appropriate.

That being said, as Review 2 states, with the exception of our research, no study has addressed the genetic basis of reproductive mode in the Teredinidae. Calloway and Turner (1988) treated morphospecies with different brooding modes as separate cryptic species. Our research supports this assumption, and indicates further studies are needed.

The possibility SyLTB is a result of a narrow window of fertilization in a SqLTB species is suggested by R#2. If a sequential long-term brooder female had a fertilization window of very short duration, the larvae within the gill would be of very similar, if not identical, age classes. However, an examination of larvae in a section of gill, with a microscope, would confirm which brooding mode the specimen had. SyLTBs larvae move freely in the gill lamella, whereas in SqLTBs larvae are contained ‘pouches’, a fusion of the gill lamellae. The extent of fertilization opportunities for a shipworm female is not known ‘in the wild’. Shipworm communities in natural wood are generally of high density with multiple age classes and with both genders well represented. This makes it unlikely that single fertilization events are the norm. This is an interesting and important line of research and thought, requiring a combination of morphological, genetic, and experimental studies, and is beyond the intended scope of this paper.

We believe the revisions enacted here address the suggestions, comments, and concerns of the editor and reviewers 1 and 2. The manuscript is significantly improved due to this input, which is greatly appreciated.

We look forward to hearing from you,

Best Regards,

Nancy C. Treneman

On behalf of the authors

Literature Cited

Calloway CB. Brooding in the Bivalvia (Mollusca). Doctoral Dissertation, Harvard University, Cambridge, MA. 1988.

Calloway CB, Turner RD. Brooding in the Teredinidae (Mollusca: Bivalvia). In: Thompson MF, Sarojiani R, Nagabhushanam R, editors. Marine biodeterioration: advanced techniques applicable to the Indian Ocean. Rotterdam: AA Balkema; 1988. pp. 215–226.

---

## [Editor Report · Decision Letter 2]

2 Mar 2026

Cosmopolitan no more: phylogenetics and reproductive mode reveal a global species complex in a marine mollusk (Teredinidae)

PONE-D-25-45073R2

Dear Dr. Treneman,

We’re pleased to inform you that your manuscript has been judged scientifically suitable for publication and will be formally accepted for publication once it meets all outstanding technical requirements.

Kind regards,

Marcos Rubal

Academic Editor

PLOS One

---

## [Editor Report · Acceptance letter]

PONE-D-25-45073R2

PLOS One

Dear Dr. Treneman,

I'm pleased to inform you that your manuscript has been deemed suitable for publication in PLOS One. Congratulations! Your manuscript is now being handed over to our production team.

Kind regards,

on behalf of

Dr. Marcos Rubal

Academic Editor

PLOS One